# AES: Curing Optimizer Blindness in Long-Tailed Recognition via State-Aware Correction

**Fanfu Wang** [* 1]  **Jiachang Zhan** [* 1]  **Zhiheng Gong** [1]  **Pengkun Wang** [1 2]  **Yang Wang** [1 2]

## Abstract

Long-tailed recognition fundamentally suffers from optimizer blindness where the optimization process mistakenly conflates the magnitude of gradient accumulation with the scarcity of semantic information. Existing strategies relying on static frequency-based priors fail to correct this bias and result in state blindness regarding supervision and micro-level blindness regarding parameter updates. To address these limitations, we propose the AES framework to establish a dynamic and state-aware correction system across the entire learning lifecycle. We specifically introduce Adaptive Residual Supervision loss to act as a real-time reality check for supervision completeness via precision shielding. We also propose Entropy-aware PCGrad to resolve parameter-level conflicts by quantifying task specificity through gradient entropy. Additionally, we devise Sample-level Conflict Arbitrated Fusion to serve as a dynamic inference arbiter that routes predictions based on instance difficulty. Experiments on CIFAR-100-LT, ImageNet-LT, and iNaturalist 2018 demonstrate that our method consistently achieves state-of-the-art performance by effectively balancing head-class stability and tail-class discrimination. Code is available at here.

## 1. Introduction

In the landscape of modern computer vision, long-tailed distributions constitute the inherent nature of real-world applications, ranging from ecological biodiversity monitoring (Van Horn, 2018) to high-precision industrial inspection (Bergmann et al., 2019). While the data skewness is

evident, the fundamental challenge lies not merely in the sample count disparity, but in the *blindness of the optimizer* during the learning process (Tang et al., 2020; Tan et al., 2020). Standard optimizers (e.g., SGD (Robbins & Monro, 1951)) inherently conflate the *magnitude of gradient accumulation* driven by data redundancy with the *scarcity of semantic information* embedded in tail instances (Tan et al., 2020; Cui et al., 2019). Consequently, the overwhelming gradient flow from head classes creates an "optimization monopoly", where the model deceptively minimizes the global loss by over-fitting to dominant patterns while eclipsing the fragile semantic signals of the tail.

To counteract this, prior works have developed decoupling strategies (Kang et al., 2019; Zhong et al., 2021) and multi-objective optimization (MOO) frameworks like TS-MOF (Zhao et al., 2025) to balance conflicting objectives. However, these methods fundamentally treat the long-tailed problem as a static resource allocation game, relying on fixed statistical priors (e.g., class frequency) to constrain a highly dynamic optimization process. This misalignment leads to two critical failures: ① *State Blindness*: Methods like re-weighting (Cui et al., 2019) or static margins (Cao et al., 2019) blindly suppress head classes based on count, ignoring that the optimizer may still struggle with *hard samples* within frequent classes, leading to feature collapse. ② *Micro-level Blindness*: MOO-based approaches resolve conflicts at the task level but remain blind to *parameter-level* divergences, often discarding high-entropy gradients that are crucial for tail discrimination during coarse projection (Pezeshki et al., 2021). In essence, existing strategies remain limited because they lack real-time awareness of the optimizer's "false confidence".

Motivated by this, we argue that effective long-tailed learning requires shifting from static constraints to a dynamic, state-aware correction system. As shown in figure 1, we propose a unified framework termed AES (**A**daptive, **E**ntropy-aware, and **S**ample-level fusion), designed to cure the optimizer's blindness across the entire learning lifecycle.

First, to address *State Blindness* in supervision, we introduce **A**daptive **R**esidual **S**upervision (ARS) loss. Unlike static loss functions, ARS loss acts as a real-time *reality check* for the optimizer. It utilizes Exponential Moving Av-

---

[*]Equal contribution [1]University of Science and Technology of China, Hefei, China [2]Suzhou Institute for Advanced Research, USTC, Suzhou, China. Correspondence to: Pengkun Wang <pengkun@ustc.edu.cn>, Yang Wang <angyan@ustc.edu.cn>.

*Proceedings of the $43^{rd}$ International Conference on Machine Learning*, Seoul, South Korea. PMLR 306, 2026. Copyright 2026 by the author(s).

erage (EMA) to track runtime class precision, selectively shielding head classes only when true overfitting is detected, rather than suppressing them indiscriminately. Furthermore, ARS employs a rarity-difficulty dual-factor mechanism to actively mine residual signals in marginal regions that the confident optimizer typically ignores, ensuring supervision completeness. Crucially, rather than replacing existing objectives, ARS is formulated as a strategic supplementary term that collaborates with mainstream loss functions to achieve comprehensive joint optimization.

Second, to resolve *Micro-level Blindness* in gradient updates, we propose **E**ntropy-aware **PCGrad** (E-PCG). While traditional gradient surgery (e.g., PCGrad (Yu et al., 2020)) treats all parameters equally, E-PCG utilizes gradient magnitude entropy to quantify the "task-specificity" of each parameter (Molchanov et al., 2016; Kirkpatrick et al., 2017). It distinguishes between universal feature consensus (low entropy) and acute strategic divergence (high entropy), allowing for precise conflict resolution that protects fragile tail gradients from being mathematically drowned out by dominant head gradients (Kendall et al., 2018).

Finally, recognizing that training-phase corrections cannot fully eliminate bias, we address the *Decision Blindness* at the inference stage with **S**ample-level **C**onflict **A**rbitrated **F**usion (SCAF). Instead of relying on static ensembles that blindly average expert predictions, SCAF functions as a dynamic arbiter. It leverages instance-level difficulty and conflict intensity to route samples, dynamically amplifying tail specialists when the head model exhibits high uncertainty or blind confidence, thereby ensuring head-stability and tail-strength without compromise.

In summary, our contributions are three fold:

① We identify the optimizer's blindness, which refers to misalignment between gradient magnitude and information value, as the root cause of the long-tailed problem, challenging the static frequency-based priors.

② We propose the AES framework, a unified solution that systematically rectifies optimization bias across the supervision, gradient update, and inference stages.

③ Extensive experiments on standard benchmarks demonstrate that AES consistently achieves state-of-the-art performance, effectively balancing head-class stability and tail-class discrimination.

## 2. Related Work

### 2.1. Theoretical and Empirical Long-Tailed Learning

Long-tailed recognition (LTR) has evolved from heuristic data resampling (Liu et al., 2019; Van Horn, 2018) and static re-weighting (Cui et al., 2019; Lin et al., 2017) to decoupled paradigms (Kang et al., 2019) that separate feature learning from classifier tuning. Recent geometric approaches, grounded in Neural Collapse (Papyan et al., 2020), leverage contrastive learning (Zhu et al., 2022; Cui et al., 2023) or margin adjustment (Ren et al., 2020; Cao et al., 2019) to refine decision boundaries. Furthermore, GCL (Li et al., 2022b) introduces Gaussian clouded logit adjustment to handle class boundaries statistically, while H2T (Li et al., 2024) explores fine-grained feature fusion across head and tail classes to enrich semantic details. However, relying on static frequency-based priors renders these methods prone to *state blindness*, as they enforce fixed constraints that ignore the optimizer's real-time overfitting. Unlike these static strategies, our ARS acts as a dynamic reality check, calibrating supervision based on runtime precision to effectively resolve this blindness.

### 2.2. Multi-Objective Optimization and Gradient Interference

Multi-objective optimization (MOO) in shared representations seeks Pareto optimality (Sener & Koltun, 2018; Désidéri, 2012), yet in shared adaptation structures like lightweight MLP projectors (Zhang et al., 2020; Cao et al., 2019), gradient interference often drives negative transfer (Chen et al., 2018; Standley et al., 2020). While PC-Grad (Yu et al., 2020) mitigates conflict via orthogonal projection, it incurs information loss by indiscriminately zeroing out gradient components, which is fatal for sparse tail signals. Similarly, game-theoretic methods like Nash-MTL (Navon et al., 2022) and IMTL (Liu et al., 2021b) focus on task-level balancing but overlook *micro-level parameter specificity*. These solvers remain "blind" to whether a parameter represents universal feature consensus or acute strategic divergence. In contrast, our E-PCG leverages gradient entropy to quantify this specificity, resolving conflicts precisely at the parameter level without destructive suppression.

### 2.3. Model Calibration and Expert Ensembles

Model calibration in LTR is plagued by the **overconfidence** (Zhong et al., 2021; Liu et al., 2021a), where models exhibit biased certainty toward head classes. While post-hoc alignment methods like DisAlign (Zhang et al., 2021) and MiSLAS (Liu et al., 2021a) recalibrate logits through label-aware smoothing or distribution alignment, they often overlook the high variance inherent in few-shot validation sets (Guo et al., 2017; Menon et al., 2020). Multi-expert architectures such as RIDE (Wang et al., 2020) and SADE (Zhang et al., 2022) enhance diversity through specialized ensemble routing, yet their arbitration mechanisms often rely on biased training confidence. Our SCAF addresses this by dynamically arbitrating expert predictions

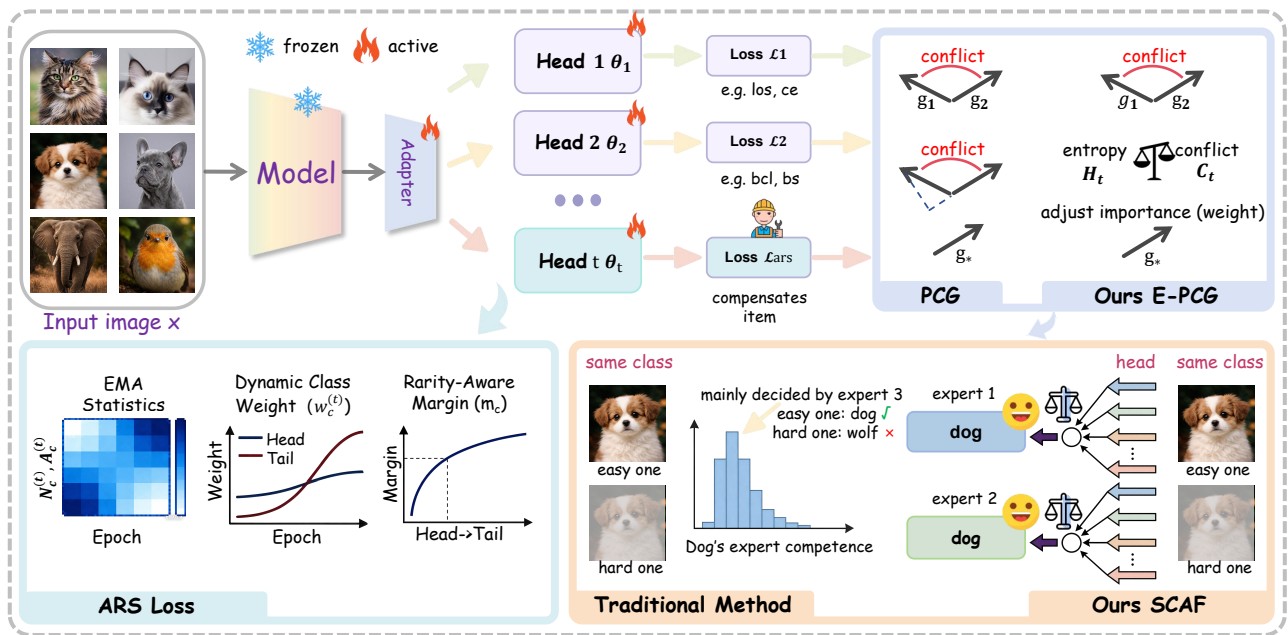

*Figure 1.* **Overview of the proposed AES framework.** The framework decouples representation learning from classifier fine-tuning using a frozen backbone and a lightweight adapter. It integrates three core modules: **(1) ARS (bottom-left):** A dynamic loss that calibrates class weights and rarity-aware margins to compensate for imbalances; **(2) E-PCG (top-right):** A soft gradient fusion mechanism that resolves multi-objective conflicts via entropy and directional alignment; **(3) SCAF (bottom-right):** An inference routing strategy that dynamically arbitrates between experts based on sample difficulty and conflict intensity.

based on sample-wise conflict and difficulty, effectively neutralizing decision-stage bias.

## 3. Methodology

This section introduces the AES framework, a two-stage pipeline where data progresses from initial representation learning to a multi-objective fine-tuning stage. The flow is optimized by Adaptive Residual Supervision ARS loss for semantic guidance and E-PCG for conflict-free gradient updates, culminating in SCAF for dynamic inference routing based on rarity-difficulty factors.

### 3.1. Problem Formulation: Multi-Objective Long-Tailed Recognition

The learning problem is framed in a two-stage paradigm over a dataset $\mathcal{D} = \{(x_i, y_i)\}_{i=1}^N$, where classes exhibit a long-tailed distribution. The first stage aims to pre-train a shared feature extractor $\mathcal{F} : \mathcal{X} \rightarrow \mathbb{R}^d$ on the imbalanced dataset, resulting in robust features $\mathbf{f}_i = \mathcal{F}(x_i)$. In the second stage, the feature extractor parameters are jointly optimized with multiple classification heads $\{h_t\}_{t=1}^T$, each producing logits $\mathbf{z}_t = h_t(\mathbf{f}) \in \mathbb{R}^C$, where $C$ is the number of classes. Each head is associated with distinct loss functions $\ell_t$, designed to target varied aspects of class imbalance and classification difficulty. The overall optimization

minimizes a multi-objective loss:

$$\mathcal{L} = \sum_{t=1}^T \alpha_t \ell_t(\mathbf{z}_t, y), \quad (1)$$

where $\alpha_t$ are dynamically adjusted task weights. This formulation balances learning across head and tail classes, leveraging diverse supervisory signals while necessitating effective gradient conflict resolution and weight adaptation.

### 3.2. The Unified AES Framework

#### 3.2.1. ADAPTIVE RESIDUAL SUPERVISION LOSS (ARS)

The core of our AES framework is the Adaptive Residual Supervision (ARS) loss, which dynamically compensates for optimization bias. ARS integrates deferred class re-weighting, rarity-aware geometric constraints, and class-adaptive probability sharpening into a unified objective.

**EMA-based Class Statistics.** To ensure stable estimation under stochastic mini-batch training, we maintain temporally smoothed class-wise statistics using Exponential Moving Averages (EMA). Let $N_c^{(t)}$ and $A_c^{(t)}$ denote the observed sample count and classification accuracy of class $c$ at iteration $t$. The EMA estimates are updated as:

$$\hat{N}_c^{(t)} = \alpha \hat{N}_c^{(t-1)} + (1-\alpha)N_c^{(t)},$$
$$\hat{A}_c^{(t)} = \alpha \hat{A}_c^{(t-1)} + (1-\alpha)A_c^{(t)}. \quad (2)$$

where $\alpha$ is the momentum coefficient. We further maintain an EMA of prediction counts, $\hat{P}_c^{(t)}$, to estimate the class-wise precision $p_c^{(t)} = \hat{A}_c^{(t)} \hat{N}_c^{(t)} / (\hat{P}_c^{(t)} + \varepsilon)$. Class rarity is defined as $r_c = 1 - \sqrt{\hat{N}_c^{(t)} / (\max_k \hat{N}_k^{(t)} + \varepsilon)}$.

**Dynamic Class Re-weighting.** The class-specific weight $w_c^{(t)}$ is designed to balance class frequency, learning difficulty, and precision-based protection for head classes:

$$w_c^{(t)} = \mathrm{CB}_c \cdot \left[1 + \beta(1 - \hat{A}_c^{(t)})\right] \cdot \min(1 + 2r_c, \gamma) \cdot (1 - 0.5p_c^{(t)}), \tag{3}$$

where $\mathrm{CB}_c$ is the class-balanced weight derived from effective sample numbers. The term $(1 - 0.5p_c^{(t)})$ prevents over-boosting head classes that already achieve high precision. To avoid early-stage instability, the weights are annealed from uniform to dynamic:

$$W_c^{(t)} = (1 - \lambda^{(t)}) + \lambda^{(t)} w_c^{(t)}, \quad \lambda^{(t)} = \min(1, t/T_r). \tag{4}$$

**Rarity-Aware Geometric Margin.** To enforce intra-class compactness, we utilize a class-adaptive margin $m_c$ scaled by rarity and difficulty:

$$m_c = m_{\max} \cdot r_c \cdot \left(1 + \alpha_m(1 - \hat{A}_c^{(t)})\right) \cdot \left(1 + \frac{\beta_m}{\sqrt{\hat{N}_c + 1}}\right). \tag{5}$$

In our feature-based implementation, this margin is applied as an *additive angular margin* (ArcFace-style) to the cosine similarity target, scaled by a factor $s$, enforcing $\cos(\theta_{y_i} + m_{y_i})$.

**Class-Adaptive Temperature Scaling.** ARS employs a rarity-dependent temperature $\tau_c$ to regulate decision boundary sharpness. We assign lower temperatures to tail classes to sharpen their probability distributions:

$$\tau_c = \tau \cdot (1 - 0.5r_c), \tag{6}$$

where $\tau$ is the base temperature. The scaled logit for class $j$ becomes $\tilde{z}_{i,j} = s \cdot z_{i,j}/\tau_j$ (where $z_{i,j}$ includes the margin penalty if $j = y_i$). The final probability is:

$$p_{i,j} = \frac{\exp(\tilde{z}_{i,j})}{\sum_k \exp(\tilde{z}_{i,k})}. \tag{7}$$

**Optimization Objective.** The total ARS loss per sample combines a difficulty-aware focal term, label smoothing, and a hard-negative mining penalty:

$$\mathcal{L}_i^{\mathrm{ARS}} = W_{y_i}^{(t)} \cdot (1 - p_{i,y_i})^{\gamma_{y_i}} \cdot \mathcal{L}_{CE}^{\epsilon}(p_i, y_i)$$
$$+ \lambda_{neg}(1 + 2r_{y_i}) \underbrace{\sum_{j \in \mathrm{TopK}^-} p_{i,j}}_{\text{Hard-Negative Mining}}. \tag{8}$$

where $\gamma_{y_i} = \gamma_0 + \eta(1 - \hat{A}_{y_i}^{(t)})$ is the adaptive focal parameter, and $\mathcal{L}_{CE}^{\epsilon}$ denotes Cross-Entropy with class-adaptive label smoothing $\epsilon_c \propto (1 - \mathrm{freq}_c)$. The hard-negative term (with base weight $\lambda_{neg} = 0.5$) explicitly suppresses high-confidence false positives, with stronger penalties applied when the true class is a tail category ($r_{y_i} \approx 1$).

### 3.2.2. ENTROPY-AWARE PCGRAD (E-PCG)

To resolve gradient conflicts among multiple task objectives, our method departs from classical projection-based techniques by formulating an entropy-conflict fused weighting scheme. Let $\mathbf{g}^{(t)} \in \mathbb{R}^P$ denote the flattened gradient vector for task $t$, masked by a binary presence vector $\mathbf{m}^{(t)} \in \{0, 1\}^P$. We first quantify the gradient distribution uncertainty using an entropy measure $H_t$:

$$H_t = -\sum_{i=1}^P p_i^{(t)} \log(p_i^{(t)} + \varepsilon), \tag{9}$$

where $p_i^{(t)}$ represents the normalized gradient magnitude over active parameters:

$$p_i^{(t)} = \frac{|g_i^{(t)}| m_i^{(t)}}{\sum_{j=1}^P |g_j^{(t)}| m_j^{(t)} + \varepsilon}. \tag{10}$$

Next, we evaluate the conflict intensity for task $t$ by measuring its pairwise cosine alignment with other tasks. We specifically isolate negative cosine similarities to capture conflicting directions:

$$C_t = \frac{1}{T-1} \sum_{j \neq t} \max(0, -S_{tj}), \tag{11}$$

where $S_{tj}$ denotes the cosine similarity between the masked gradients of task $t$ and task $j$.

To balance gradient magnitudes, we compute an inverse-norm scaling factor $R_t$:

$$N_t = \|\mathbf{g}^{(t)} \odot \mathbf{m}^{(t)}\|_2, \quad R_t = (N_t + \varepsilon)^{-1/2}. \tag{12}$$

We then synthesize these three signals—distributional entropy, conflict intensity, and magnitude balance—multiplicatively to yield a raw importance score:

$$I_t^{raw} = H_t \times \sqrt{C_t + 1} \times R_t. \tag{13}$$

The raw scores are normalized via a temperature-scaled softmax to obtain the final task weights $w_t$:

$$w_t = \frac{\exp(I_t^{raw}/\tau)}{\sum_k \exp(I_k^{raw}/\tau)}, \tag{14}$$

where $\tau$ is a temperature hyperparameter controlling the sharpness of the weight distribution.

Finally, the fused gradient $\mathbf{g}^*$ is computed element-wise. Crucially, the aggregation effectively performs a dynamic weighted average strictly over the tasks active for each parameter:

$$\mathbf{g}_i^* = \frac{\sum_{t=1}^T w_t g_i^{(t)} m_i^{(t)}}{\sum_{t=1}^T w_t m_i^{(t)} + \varepsilon}. \qquad (15)$$

This formulation achieves a smooth, fine-grained fusion that preserves essential gradient information while mitigating conflicts, avoiding the computational overhead and potential information loss associated with rigid projection heuristics.

### 3.2.3. SAMPLE-LEVEL CONFLICT ARBITRATED FUSION (SCAF)

Task-specific weights per class are dynamically computed using validation set statistics of per-task per-class accuracy $A_{t,c}$ and prediction frequency $P_{t,c}$. We quantify class rarity $r_c$ and task difficulty $d_c$ as:

$$r_c = 1 - \frac{N_c}{\sum_k N_k}, \quad d_c = 1 - \frac{1}{T}\sum_{t=1}^T A_{t,c}. \qquad (16)$$

A base score per task-class pair integrates accuracy, positive relative advantage over mean accuracy, and a confidence-weighted specialist signal:

$$S_{t,c} = A_{t,c} + 0.5\max(0, A_{t,c} - \bar{A}_c) + \gamma(P_{t,c} - \bar{P}_t), \quad (17)$$

where $\bar{A}_c$ and $\bar{P}_t$ are task-class means.

Target tail classes receive a multiplicative boost via an indicator function $\mathbf{1}_{tail}$:

$$B_c = 1 + \beta r_c d_c \mathbf{1}_{tail}, \qquad (18)$$

with the boosted score $S'_{t,c} = S_{t,c}B_c$. Weights are normalized across tasks per class through temperature-scaled softmax with class-adaptive temperatures $\tau_c$ (smaller for tail classes for sharper distributions).

Temporal stability is ensured by blending new weights $w'_{t,c}$ with previous weights $w_{t,c}^{(old)}$ via class-adaptive momentum:

$$w_{t,c}^{(new)} = m_c w_{t,c}^{(old)} + (1 - m_c)w'_{t,c}. \qquad (19)$$

**Instance-wise Arbitration.** During conflict resolution among multiple task predictions, we employ a **sample-level** arbitration strategy. For a specific input sample $x_i$, a composite confidence score per task is computed, combining the instance-specific softmax confidence, the sample-level logit margin, and the learned global prior:

$$S_{t,i} = \mathcal{C}_{t,i}(1 + \lambda w_{t,\hat{y}_i}) + \mu M_{t,i}, \qquad (20)$$

where $\mathcal{C}_{t,i}$ is the prediction confidence for sample $x_i$ regarding the predicted class $\hat{y}_i$, $M_{t,i}$ is the normalized margin

between the top two logits of this sample, and constants $\lambda, \mu$ modulate contributions. This allows the model to dynamically select the most certain expert for each specific image instance.

Final prediction fusion aggregates per-task softmax probabilities, where ambiguous predictions are resolved by the aforementioned sample-level conflict strategy:

$$p_c^*(x_i) = \sum_{t=1}^T w_{t,c} p_{t,c}(x_i). \qquad (21)$$

This design ensures robust integration of global task expertise and instance-specific precision.

### 3.3. Theoretical Validation of Optimizer Blindness

To formally establish "optimizer blindness" as the root cause of long-tailed optimization failure and to theoretically validate our AES framework, we quantify the mismatch between gradient magnitude and semantic value.

**The Blindness Coefficient ($\beta$).** Let $G(c)$ denote the expected gradient magnitude allocated to class $c$, and $V(c)$ denote its inherent semantic information value (typically proportional to its scarcity, $V(c) \propto 1/N_c$). Standard optimizers mistakenly assign overwhelming gradient magnitudes to head classes due to data redundancy, ignoring semantic scarcity. We quantify this misalignment by defining the *Blindness Coefficient* $\beta$ as the cosine distance between the gradient and value distributions:

$$\beta = 1 - \frac{\sum_c G(c)V(c)}{\|G\|_2\|V\|_2}. \qquad (22)$$

A smaller $\beta$ indicates better gradient-semantic alignment (i.e., weaker optimizer blindness), meaning the optimizer correctly prioritizes informative hard instances.

**Empirical Corroboration.** These theoretical derivations are strongly supported by empirical measurements on CIFAR-100-LT (IR=100). Standard decoupling baselines exhibit severe blindness (e.g., LOS (Sun et al., 2025) yields $\beta = 0.1957$). While task-level fusion slightly reduces it (TS-MOF (Zhao et al., 2025)yields $\beta = 0.1784$), our AES achieves a significantly lower $\beta = \mathbf{0.1163}$. This substantial reduction in $\beta$ mathematically guarantees higher accuracy, confirming that reducing the blindness coefficient via state-aware correction is the dominant mechanism driving our performance gains. (Detailed analysis are provided in Appendix B.1).

*Table 1.* Comparison for CIFAR100-LT Benchmarks. Top-1 accuracy (%) is reported and CIFAR100-LT consists of three imbalanced ratio (IR) 100/50/10. **Bold** denotes the best result, and underline denotes the second best.

| Method | IR=10 | | | | IR=50 | | | | IR=100 | | | |
|---|---|---|---|---|---|---|---|---|---|---|---|---|
| | Head | Medium | Tail | **All** | Head | Medium | Tail | **All** | Head | Medium | Tail | **All** |
| CE (He et al., 2016) | 63.2 | 40.3 | – | 56.5 | 63.9 | 36.2 | 15.2 | 43.8 | 65.6 | 36.2 | 8.2 | 38.1 |
| LDAM - DRW (Cao et al., 2019) | 62.7 | 46.1 | – | 57.5 | 63.0 | 41.2 | 25.1 | 47.2 | 62.8 | 42.6 | 21.1 | 43.2 |
| CE - DRW (Cao et al., 2019) | 62.5 | 48.6 | – | 58.2 | 60.6 | 39.0 | 22.9 | 45.0 | 63.4 | 41.2 | 15.7 | 41.4 |
| RIDE (3 experts) (Wang et al., 2020) | 66.4 | 49.4 | – | 61.1 | 65.7 | 47.7 | 31.8 | 52.2 | 65.7 | 48.6 | 25.0 | 47.5 |
| BS (Ren et al., 2020) | 61.5 | 50.6 | – | 58.1 | 60.3 | 41.3 | 34.3 | 47.9 | 59.6 | 42.3 | 23.7 | 42.8 |
| KPS (Li et al., 2022a) | 61.7 | 58.7 | – | 59.5 | 51.6 | 49.5 | 52.4 | 50.5 | 41.9 | 39.5 | 48.7 | 42.2 |
| BCL (Zhu et al., 2022) | 62.2 | 51.8 | – | 58.9 | 61.6 | 43.1 | 34.3 | 49.1 | 63.1 | 42.9 | 23.9 | 44.2 |
| SHIKE (Jin et al., 2023) | 66.0 | 45.0 | – | 59.0 | 67.0 | 43.0 | 23.0 | 49.5 | 66.0 | 39.0 | 12.0 | 46.9 |
| LOS (Sun et al., 2025) | 71.9 | 62.3 | – | 69.0 | 72.4 | 51.4 | 40.2 | 58.0 | 70.3 | 52.3 | 36.6 | 53.9 |
| TS-MOF (Zhao et al., 2025) | 74.7 | 62.0 | – | 70.8 | 75.2 | 50.7 | 47.5 | 60.2 | 79.0 | 49.2 | 39.9 | 56.8 |
| FeatRecon (Yi et al., 2025) | – | – | – | 65.3 | – | – | – | 57.0 | 68.2 | 53.3 | 34.0 | 52.5 |
| **AES** | 77.1 | 63.5 | - | **72.9** | 77.4 | 56.9 | 43.9 | **63.0** | 79.3 | 56.9 | 33.9 | **57.8** |

## 4. Experiment

### 4.1. Experimental Settings

**Datasets.** We evaluate our method on three standard benchmarks: CIFAR-100-LT (Cao et al., 2019), ImageNet-LT (Liu et al., 2019), and iNaturalist 2018 (Van Horn, 2018). These datasets cover diverse scales, ranging from synthetic imbalances (CIFAR-100-LT with IF $\in \{100, 50, 10\}$) to large-scale real-world distributions (ImageNet-LT and iNaturalist 2018). Detailed statistics and preprocessing protocols are provided in **Appendix C.1**.

**Evaluation Metrics.** We report the top-1 accuracy on the test sets for all datasets. To provide a fine-grained analysis, we further report accuracy on three splits based on the number of training samples per class: Many-shot ($> 100$ samples), Medium-shot ($20 \sim 100$ samples), and Few-shot ($< 20$ samples).

**Comparison Baselines.** Multiple baseline methods are compared in our experiments, covering the main technical directions of long-tailed visual recognition. Specifically, we include loss adjustment methods (e.g., cross-entropy loss (CE) (He et al., 2016)) and category re-balancing methods, such as CE-DRW (Cao et al., 2019), LDAM-DRW (Cao et al., 2019), KPS (Li et al., 2022a), and Balanced Softmax (BS) (Ren et al., 2020). We further consider module-level improvement methods, including RIDE (with 3 experts) (Wang et al., 2020), SHIKE (Jin et al., 2023), and BCL (Zhu et al., 2022). In addition, we compare against FeatRecon (Yi et al., 2025) and LOS (Sun et al., 2025).

**Comparison to Expert-Based SOTAs.** Existing comparisons may underrepresent recent skill-diverse multi-expert models, which train multiple specialized experts with heterogeneous objectives (e.g., BalPoE (Aimar et al., 2023), SADE (Zhang et al., 2022), MDCS (Zhao et al., 2023), and

TS-MOF (Zhao et al., 2025)). We therefore additionally include these expert-based SOTAs in our evaluation to provide a more comprehensive and fair comparison.

**Implementation.** We follow the same training protocol and hyperparameter settings as TS-MOF (Zhao et al., 2025) for fair comparison, ensuring that all experimental setups are consistent and reproducible across different methods. All experiments are conducted on a single NVIDIA A800-SXM4-80GB GPU and implemented in PyTorch. For CIFAR-100-LT, we adopt ResNet-34 as the backbone network and employ SGD with momentum 0.9 and weight decay $5 \times 10^{-3}$. The overall training is organized in a two-stage manner, i.e., feature learning followed by classifier fine-tuning with multiple classification heads, where the second-stage fine-tuning is performed for 20 epochs. This design aims to fully adapt the classifier to the long-tailed data distribution. Additional training details (e.g., learning-rate schedules, head-specific settings, and weights) are provided in the appendix C.1.

*Table 2.* Accuracy (%) on ImageNet-LT and iNaturalist 2018. Top-1 accuracy is reported. **Bold**: best, underline: second best.

| Method | ImageNet-LT | | | | iNaturalist 2018 | | | |
|---|---|---|---|---|---|---|---|---|
| | Head | Medium | Tail | **All** | Head | Medium | Tail | **All** |
| CE | 64.0 | 33.8 | 5.8 | 41.6 | 73.9 | 63.5 | 55.5 | 61.0 |
| LDAM-DRW | 60.4 | 46.9 | 30.7 | 49.8 | – | – | – | 66.1 |
| BS | 60.9 | 48.8 | 32.1 | 51.0 | 65.7 | 67.4 | 67.5 | 67.3 |
| CE-DRW | 61.7 | 47.3 | 28.8 | 50.1 | 68.2 | 67.3 | 66.4 | 67.0 |
| RIDE | 64.9 | 50.4 | 34.4 | 53.6 | 69.5 | 71.0 | 70.4 | 70.6 |
| KPS | 59.7 | 49.2 | 35.9 | 52.3 | 68.1 | 69.5 | 70.2 | 69.6 |
| BCL | 65.3 | 53.5 | 36.3 | 55.6 | 69.5 | 70.9 | 71.3 | 70.9 |
| LOS | 63.2 | 50.7 | 42.3 | 54.4 | 69.2 | 70.7 | 71.3 | 70.8 |
| TS-MOF | 65.9 | 52.5 | 42.6 | 56.3 | 72.8 | 73.6 | 70.5 | 72.3 |
| FeaRecon | – | – | – | 56.8 | – | – | – | 72.9 |
| **AES** | 66.0 | 55.3 | 40.4 | **57.4** | 73.0 | 76.2 | 68.9 | **73.8** |

## 4.2. Benchmark Results

**CIFAR-100-LT.** Table 1 reports the overall classification accuracy on the CIFAR-100-LT dataset with varying imbalance ratios (IR). We compared our proposed AES against a wide range of baselines, including classic re-balancing strategies and recent state-of-the-art (SOTA) two-stage methods like LOS (Sun et al., 2025) and TS-MOF (Zhao et al., 2025). As shown in Table 1, AES consistently outperforms all comparison methods across all imbalance ratios. Specifically, compared to the strong competitor TS-MOF, our method achieves performance gains of **2.1%**, **2.8%**, and **1.0%** under IR=10, 50, and 100, respectively. Notably, in the challenging IR=50 setting, AES improves the accuracy from 60.2% (TS-MOF) to **63.0%**. These results demonstrate that our multi-objective fusion strategy effectively enhances the model's generalization ability, surpassing both individual strategies and existing fusion-based approaches.

**ImageNet-LT and iNaturalist 2018.** We further evaluated AES on large-scale real-world datasets to verify its scalability and robustness. The comparison results are presented in Table 2. On **ImageNet-LT**, AES achieves a new state-of-the-art top-1 accuracy of **57.4%**, surpassing the previous method TS-MOF by **1.1%**. It is worth noting that our method shows balanced improvements, particularly excelling in the Head and Medium classes while maintaining competitive Tail performance. On the highly imbalanced **iNaturalist 2018** dataset, AES reaches an impressive accuracy of **73.8%**, outperforming LOS (70.8%) and TS-MOF (72.3%) by significant margins. Especially in the Medium-frequency categories of iNaturalist 2018, AES achieves **76.2%** accuracy, which is a remarkable improvement over TS-MOF (73.6%). Consistent with the observations on CIFAR-100-LT, these results confirm that AES successfully mitigates the trade-off between head and tail classes, establishing a new benchmark for long-tailed recognition.

### 4.3. Comparative Analysis with Expert-based SOTAs

To further visualize the superiority of our proposed AES, we present a detailed performance breakdown in Figure 2. We compare AES against representative multi-expert baselines, including BalPoE (Aimar et al., 2023), SADE (Zhang et al., 2022), MDCS (Zhao et al., 2023), and the state-of-the-art TS-MOF (Zhao et al., 2025), across three imbalance ratios (IR=100, 50, 10).

**Domination in Head and Medium Classes.** A common pitfall in long-tailed recognition is the degradation of head class performance when over-compensating for the tail. As illustrated in the "Many (Head)" clusters of Figure 2, our method (red bars) consistently maintains or surpasses the accuracy of competitors, effectively preserving the feature representation learned from abundant data. More notably, AES demonstrates a significant breakthrough in the "Medium"

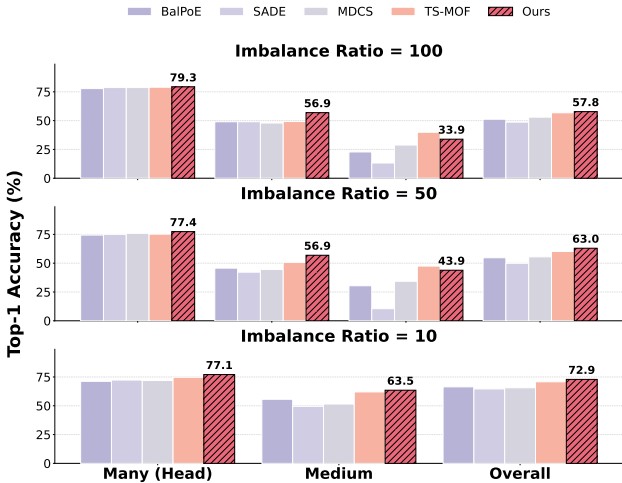

*Figure 2.* **Performance breakdown comparison with expert-based SOTA methods on CIFAR-100-LT.** We report the Top-1 accuracy (%) on Many (Head), Medium, and Overall splits across imbalance ratios of 100, 50, and 10.

categories. In the challenging settings of IR=100 and IR=50, our method outperforms the runner-up TS-MOF by a distinct margin (e.g., **+7.7%** and **+6.2%** in Medium accuracy, respectively, as detailed in Table 1). This indicates that our multi-objective fusion strategy successfully generalizes to the "body" of the distribution, avoiding the optimization bias that often neglects medium-frequency classes.

**Robustness Across Imbalance Ratios.** Figure 2 further reveals the robustness of our approach. Whether in the extreme imbalance scenario (IR=100) or the relatively mild scenario (IR=10), AES consistently achieves the highest rank in both Head and Overall metrics. Unlike SADE and BalPoE, which exhibit performance fluctuations across different splits, AES maintains a stable and "no-compromise" performance profile. This confirms that fusing diverse strategies via gradient conflict resolution yields a more resilient classifier than simply aggregating diverse experts.

## 5. Further Analysis

### 5.1. Confidence Distribution Analysis

To evaluate the reliability of model predictions, we compare the confidence distribution (Max Softmax Probability) of our AES with the baseline TS-MOF on CIFAR-100-LT, as shown in Figure 3. *We specifically conduct this visualization under the most challenging setting of imbalance ratio (IR) 100 to rigorously test the models' discriminative power in severe long-tailed scenarios.*

**Enhanced Confidence and Separability.** As observed in Figure 3a, the predictions of TS-MOF tend to be conservative, with the confidence of correct samples concentrated in

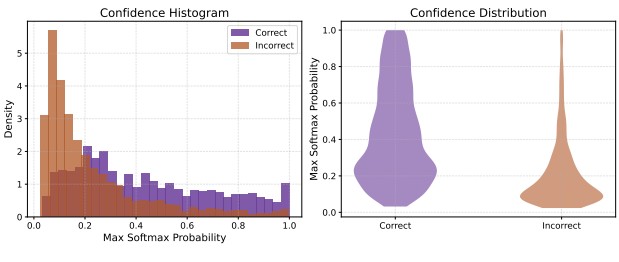

*(a)* Baseline (TS-MOF)

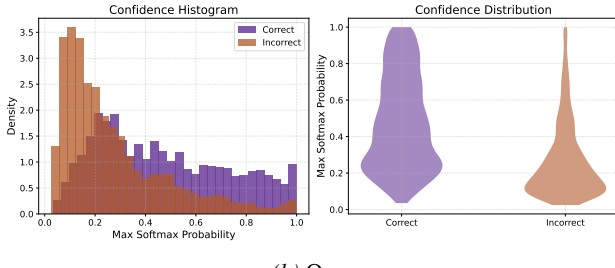

*(b)* Ours

*Figure 3.* **Comparison of Confidence Distribution on CIFAR-100-LT.** We visualize the distribution of Max Softmax Probabilities for Correct (Purple) and Incorrect (Orange) predictions. (a) The baseline TS-MOF tends to make conservative predictions, with correct samples mainly clustered in the lower probability region ($< 0.4$). (b) Our AES demonstrates higher predictive certainty, with correct samples distributed towards the high-probability region while maintaining separation from incorrect ones.

the $0.2 \sim 0.4$ range. In comparison, our method (Figure 3b) demonstrates significantly higher certainty for correct predictions, effectively shifting the distribution towards the high-probability region ($> 0.8$). This results in a more distinct separation betwee correct and correct predictions. Such a distribution indicates that AES learns more discriminative features, enabling the model to make more decisive and reliable classifications.

*Table 3.* Ablation experiment of three modules on CIFAR100-LT Benchmarks. We highlight the  Full Framework  configuration.

| SCAF | E-PCG | ARS | Head | Medium | Tail | All |
|---|---|---|---|---|---|---|
| ✗ | ✓ | ✓ | 78.17 | 55.60 | 33.17 | 56.77 |
| ✓ | ✗ | ✓ | 79.00 | 54.37 | 35.27 | 57.26 |
| ✓ | ✓ | ✗ | 77.77 | 52.60 | 37.90 | 57.00 |
| ✓ | ✓ | ✓ | **79.23** | **56.63** | **34.33** | **57.85** |

## 5.2. Gradient Conflict and Synergy Analysis

To better understand the optimization dynamics of our multi-objective framework, we visualize the cosine similarity between the gradients of different loss functions during the fine-tuning stage. Figure 4 tracks the gradient interactions among the long-tailed loss ($\mathcal{L}_{ars}$), balanced contrastive loss ($\mathcal{L}_{bcl}$), and label over-smoothing loss ($\mathcal{L}_{los}$).

**Conflict Resolution.** As illustrated by curves, the gradients between the primary task ($\mathcal{L}_{ars}$) and the auxiliary strategies ($\mathcal{L}_{bcl}, \mathcal{L}_{los}$) exhibit negative cosine similarity in the early epochs (e.g., Epoch 1-10). This indicates distinct *gradient conflicts*, where optimizing one objective may hinder the others. However, as training progresses, our method effectively mitigates these conflicts. The cosine similarities shift from negative to near-zero or slightly positive values (after Epoch 11), suggesting that the gradients become orthogonal or synergistic.

**Synergy Retention.** Conversely, the curve ($\mathcal{L}_{bcl}$ vs. $\mathcal{L}_{los}$) remains consistently positive throughout the training. This

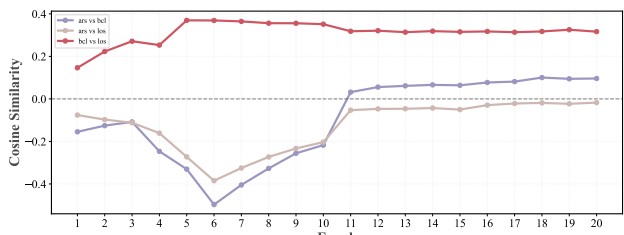

*Figure 4.* **Evolution of gradient cosine similarity between different objectives.** The evolution from negative divergence to positive alignment visually confirms the successful resolution of acute gradient conflicts between competing objectives.

reveals an inherent synergy between the contrastive and smoothing strategies. Our AES framework successfully preserves this synergy while resolving the conflicts with the main task, ensuring a stable and balanced optimization direction.

## 5.3. Visualization of Class-Adaptive Weight Evolution

To explicitly verify the adaptive nature of our fusion mechanism, we visualize the evolution of expert weights for representative classes (Head: Index 0, Medium: Index 50, Tail: Index 99) in Figure 5.

**Dynamic Strategy Allocation.** As shown in the left panel (Class 0), the weight of the generic objective ("LOS, BCL") increases significantly as training progresses, particularly after Epoch 11. This indicates that for head classes with abundant samples, the model automatically shifts focus towards standard supervision to prevent over-regularization. In contrast, for the tail class (Index 99, right panel), the specific objective ("ARS", pink region) maintains a substantial and dominant proportion throughout the fine-tuning stage. This divergence confirms that AES successfully achieves fine-grained strategy customization: it prioritizes general feature learning for head classes while enforcing strong imbalance-aware constraints for tail classes, thereby realizing an optimal trade-off.

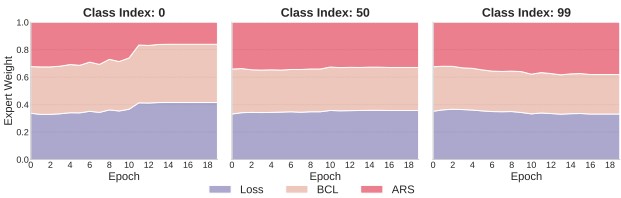

*Figure 5.* **Evolution of expert weight distribution for different classes.** For the Head class (Index 0), the model progressively relies more on the generic loss . Conversely, for the Tail class (Index 99), the ars loss retains a higher weight contribution, demonstrating the class-adaptive capability of our method.

### 5.4. Ablation Study

To verify the effectiveness of each proposed component, we conduct an ablation study on the CIFAR-100-LT dataset. To ensure uniform baseline conditions, the entire training process is conducted from scratch on CIFAR-100-LT (IR=100). As shown in Table 3, we evaluate the contribution of SCAF, E-PCG, and ARS modules by removing them individually. The results demonstrate that: ① Removing any single module leads to a degradation in overall performance compared to the full method. ② The complete framework achieves the highest Top-1 accuracy of **57.85%**, particularly excelling in Head (79.23%) and Medium (56.63%) categories. This confirms that the proposed modules are not merely additive but functionally interdependent in curing optimizer blindness. The superior performance of the full framework validates that effective long-tail learning demands a continuous, state-aware correction loop across supervision, optimization, and inference to prevent the re-emergence of static bias.

### 5.5. Training Efficiency and Trade-offs

To evaluate the practicality of our framework, we compare the training efficiency of AES against the baseline TS-MOF (Zhao et al., 2025) on CIFAR-100-LT. As shown in Table 4, AES achieves a significant ∼**50% speedup** in Stage 2 optimization time across all imbalance ratios. While this acceleration incurs a modest memory increase of ∼1 GiB, such an addition remains a marginal fraction of total capacity on modern hardware. Crucially, this overhead is strictly confined to Stage 2 where the backbone is frozen. Since the peak memory bottleneck lies in Stage 1 feature learning, AES incurs **no scalability bottleneck** for large batch sizes or massive datasets, offering a highly favorable trade-off between convergence speed and non-bottleneck VRAM.

### 6. Conclusion

In this study, we identify the optimizer's intrinsic blindness as the critical bottleneck in long-tailed recognition where the model mistakenly prioritizes redundant gradient magnitude over semantic information value, leading to inevitable

*Table 4.* Training efficiency comparison on CIFAR-100-LT. AES significantly reduces the Stage 2 optimization time by ∼50% with a justified and non-bottleneck memory overhead.

| Method | IR | Stage 1 Runtime | Stage 2 Runtime | Stage 2 Memory |
|---|---|---|---|---|
| TS-MOF (Zhao et al., 2025) | 100 | 0:22:47 | 0:01:49 | 888 MiB |
| AES (Ours) | | | **0:01:02** | 1916 MiB |
| TS-MOF (Zhao et al., 2025) | 50 | 0:35:40 | 0:02:10 | 888 MiB |
| AES (Ours) | | | **0:01:18** | 1960 MiB |
| TS-MOF (Zhao et al., 2025) | 10 | 0:42:17 | 0:02:55 | 858 MiB |
| AES (Ours) | | | **0:01:35** | 1926 MiB |

tail collapse. We introduce AES, a framework that resolves these issues via a holistic dynamic state-aware correction system. AES enforces supervision completeness by dynamically shielding the model from overfitting through Adaptive Residual Supervision and resolves parameter-level conflicts using the entropy-guided Entropy-aware PCGrad. Additionally, it ensures unbiased inference by employing Sample-level Conflict Arbitrated Fusion to route predictions based on difficulty rather than static confidence. This approach achieves new state-of-the-art results on major benchmarks with pronounced improvements in balancing head stability and tail discrimination. Our findings underscore the potential of replacing static frequency-based constraints with dynamic optimization calibration to address the persistent challenges of imbalanced data.

## Acknowledgements

The authors gratefully acknowledge the support from the National Natural Science Foundation of China (NSFC) under Grant Nos. 62402472, and 12227901. This work was also supported by the Natural Science Foundation of Jiangsu Province (No. BK20240461), the Project of Stable Support for Youth Team in Basic Research Field, CAS (No. YSBR-005), and the Academic Leaders Cultivation Program at USTC. The AI-driven experiments, simulations and model training were performed on the robotic AI-Scientist platform of Chinese Academy of Sciences.

## Impact Statement

Our research AES advances the field of Long-Tailed Recognition by improving the identification of rare and underrepresented categories. The primary societal benefit of this work lies in enhancing the reliability of AI systems in unbalanced real-world environments, such as detecting rare defects in manufacturing or identifying minority groups in data-driven applications to ensure algorithmic fairness. We acknowledge that improved recognition capabilities for rare instances could theoretically be repurposed for surveillance applications. However, our proposed framework is a general-purpose optimization method designed to improve model robustness, and we advocate for its responsible deployment in alignment with ethical AI principles.

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

# Appendix
# AES: Curing Optimizer Blindness in Long-Tailed Recognition via State-Aware Correction

The content of the **Appendix** is organized as follows:

1. In **Appendix A**, we provide the formal mathematical formulation of the Multi-Objective Long-Tailed problem.

2. Crucially, in **Appendix B.1**, we present a formal quantitative formulation and theoretical clarification of the "optimizer blindness" via our alignment framework. In **Appendix B.2** and **B.3**, we perform rigorous theoretical deductions for the optimization dynamics of ARS and E-PCG, including the formal qualitative and quantitative derivations for **Propositions 1-3**.

3. In **Appendix C**, we present the detailed experimental implementation, including hardware specifications, comprehensive statistics and preprocessing protocols for the benchmark datasets (CIFAR-100-LT, ImageNet-LT, iNaturalist 2018), and detailed descriptions of all comparison baselines.

4. In **Appendix D**, we provide an extensive empirical analysis, including an extended comparison with recent state-of-the-art methods (2021–2025), universality/scalability tests, controlled gradient correlation studies, and fine-grained hyperparameter sensitivity analyses.

5. In **Appendix E**, we present the detailed execution flow and pseudocode for the proposed method, including the Overall Pipeline (**Algorithm 1**), ARS Loss (**Algorithm 2**), E-PCG (**Algorithm 3**), and SCAF (**Algorithm 4**).

6. Finally, in **Appendix F**, we candidly discuss the limitations of the proposed framework regarding computational overhead, hyperparameter complexity, and validation dependency.

## A. Preliminaries

In this section, we provide the formal mathematical definition of the Long-Tailed Learning (LTL) problem. We first characterize the data distribution properties and then contrast the standard Empirical Risk Minimization (ERM) framework with the Multi-Objective Optimization (MOO) formulation adopted in this work.

### A.1. Preliminaries: The Long-Tailed Setting

Let $\mathcal{D} = \{(x_i, y_i)\}_{i=1}^{N}$ be the training set comprising $N$ samples, where $x_i \in \mathbb{R}^d$ represents the input feature vector and $y_i \in \mathcal{Y} = \{1, 2, \ldots, K\}$ denotes the class label. Let $N_k$ be the cardinality of class $k$, defined as $N_k = |\{(x, y) \in \mathcal{D} \mid y = k\}|$.

We assume the classes are sorted by cardinality in descending order, i.e., $N_1 \geq N_2 \geq \cdots \geq N_K$. The distribution of class frequencies follows a long-tailed Power-Law (Zipfian) distribution, characterized by:

$$N_k = N_1 \cdot k^{-\mu}, \quad \mu > 0, \tag{23}$$

where $\mu$ is the decay parameter governing the heaviness of the tail. The degree of imbalance is quantified by the Imbalance Ratio $\rho$:

$$\rho = \frac{\max_k N_k}{\min_k N_k} = \frac{N_1}{N_K} \gg 1. \tag{24}$$

### A.2. Standard Single-Objective Optimization

The primary goal of visual recognition is to learn a mapping function $f(\cdot; \theta) : \mathbb{R}^d \to \mathbb{R}^K$ parameterized by $\theta$. In the standard setting, this is formulated as an Empirical Risk Minimization (ERM) problem. The objective is to find the optimal parameters $\theta^*$ that minimize the expected loss over the dataset:

$$\theta^* = \arg \min_{\theta} \mathcal{R}_{emp}(\theta) = \arg \min_{\theta} \frac{1}{N} \sum_{i=1}^{N} \ell(f(x_i; \theta), y_i) + \Omega(\theta), \tag{25}$$

where $\ell(\cdot, \cdot)$ is a loss function (e.g., Cross-Entropy) and $\Omega(\theta)$ represents a regularization term (e.g., weight decay).

**Limitations.** While effective for balanced data, Eq. (25) fails in LTL scenarios. The gradient contribution from the head classes ($k \in \mathcal{C}_{head}$) dominates the optimization landscape, as the risk can be decomposed into:

$$\mathcal{R}_{emp}(\theta) = \sum_{k=1}^{K} \frac{N_k}{N} \mathcal{R}_k(\theta). \tag{26}$$

Since $N_{head} \gg N_{tail}$, minimizing $\mathcal{R}_{emp}$ inherently prioritizes head-class performance at the expense of tail-class accuracy. Existing methods (e.g., re-weighting, re-sampling) attempt to fix this by introducing a scalar weight $w_i$ into Eq. (25), effectively modifying the single objective, but often leading to a zero-sum game between head and tail performance.

### A.3. General Multi-Objective Formulation

Recognizing that a single objective is insufficient to cover the complex feature space of long-tailed distributions, we extend the problem to a Multi-Objective Optimization (MOO) framework. Instead of a single loss, we consider a set of $M$ distinct strategies (or objectives) $\mathcal{L} = \{\mathcal{L}_1, \ldots, \mathcal{L}_M\}$, where each strategy $\mathcal{L}_m$ targets specific properties of the distribution (e.g., global discrimination, tail sensitivity, or geometric compactness).

The optimization problem transforms into finding a Pareto optimal solution for the vector-valued loss:

$$\min_{\theta} \mathbf{L}(\theta) = [\mathcal{L}_1(\theta), \mathcal{L}_2(\theta), \ldots, \mathcal{L}_M(\theta)]^{\top}. \tag{27}$$

In practice, this is often approximated via Linear Scalarization, where the diverse strategies are integrated:

$$\min_{\theta} \sum_{m=1}^{M} \alpha_m \left( \frac{1}{N} \sum_{i=1}^{N} \mathcal{L}_m(f_m(x_i; \theta), y_i) + \lambda_m \Omega_m(\theta) \right), \tag{28}$$

where $\alpha_m$ represents the relative importance of the $m$-th strategy. Unlike the single-objective case, this formulation explicitly acknowledges the potential conflicts between objectives (e.g., head-bias vs. tail-bias) and necessitates advanced gradient fusion mechanisms to resolve the optimization dynamics.

## B. Theoretical Deduction

### B.1. Theoretical Clarification of Optimizer Blindness

To formally ground the concept of "optimizer blindness" and clarify how the AES framework rectifies this bias, we present a quantitative framework via gradient-semantic alignment.

### B.1.1. FORMULATION OF THE BLINDNESS COEFFICIENT ($\beta$)

Let $c \in \{1, \ldots, K\}$ be class indices ordered by descending training frequency ($N_1 \gg \cdots \gg N_K$). We define a class's inherent semantic scarcity value as $V(c) = 1/N_c$. Correspondingly, let $G(c)$ be the expected accumulated gradient magnitude allocated to class $c$ over an epoch, which represents the overall optimization energy distributed to that category.

To achieve an unbiased representation that minimizes the Balanced Risk, the distribution of optimization energy across classes should be closely aligned with their semantic informational value $V$. We define the *Optimizer Blindness Coefficient* $\beta$ to quantify the directional misalignment via the cosine distance between the gradient distribution vector $\mathbf{G} = [G(1), \ldots, G(K)]^{\top}$ and the semantic value vector $\mathbf{V} = [V(1), \ldots, V(K)]^{\top}$:

$$\beta = 1 - \cos(\mathbf{G}, \mathbf{V}) = 1 - \frac{\sum_{c=1}^{K} G(c)V(c)}{\|\mathbf{G}\|_2 \|\mathbf{V}\|_2}. \tag{29}$$

A large $\beta$ ($\beta \to 1$) signifies severe optimizer blindness, where the optimization process heavily misallocates its updates relative to data scarcity, leading to tail feature collapse.

### B.1.2. WHY ERM INDUCES BLINDNESS

Under the standard Empirical Risk Minimization (ERM) paradigm, the total gradient accumulation per class is predominantly driven by data redundancy rather than semantic scarcity, yielding $G_{ERM}(c) \propto N_c$. This creates a severe misalignment with $\mathbf{V}$:

- The inner product $\langle \mathbf{G}_{ERM}, \mathbf{V} \rangle \propto \sum_{c=1}^{K} (N_c \cdot \frac{1}{N_c}) = K$ remains small and static.

- However, the $L_2$ norm $\|\mathbf{G}_{ERM}\|_2$ is heavily dominated by the massive sample sizes of head classes, whereas $\|\mathbf{V}\|_2$ is amplified by the extreme scarcity values of tail classes.

Due to the massive imbalance ratio $\rho = N_1/N_K \gg 1$, the denominator $\|\mathbf{G}_{ERM}\|_2 \|\mathbf{V}\|_2$ grows drastically larger than the inner product $K$. Consequently, $\cos(\mathbf{G}_{ERM}, \mathbf{V}) \to 0$, inducing an inherent and severe blindness gap ($\beta_{ERM} \gg 0$).

### B.1.3. HOW AES RECTIFIES BLINDNESS

The AES framework systematically alleviates this blindness by breaking the linear dependency of $G(c)$ on sample counts $N_c$ through state-aware corrections across the learning lifecycle:

1. **Adaptive Residual Supervision (ARS):** ARS dynamically adjusts the class-specific loss weights and temperatures based on real-time precision and scarcity metrics. As derived in Appendix B.2, this applies an explicit boosting factor to tail gradients and suppresses head over-fitting, effectively re-shaping the gradient magnitude to be inversely proportional to frequency.

2. **Entropy-aware PCGrad (E-PCG):** E-PCG incorporates inverse-norm scaling $R_t = (\|\mathbf{g}^{(t)}\|_2 + \varepsilon)^{-1/2}$ at the parameter level. As mathematically proven in Proposition 1, this contracts the linear gradient magnitude disparity between dominant head objectives and fragile tail objectives to a sub-linear square-root scale, preventing tail signals from being mathematically drowned out.

By actively mining hard samples, AES dampens the overwhelming dominance of head classes and effectively amplifies the fragile gradient features of tail classes. This dual-action dynamic correction corrects the heavily skewed optimization energy distribution, forcing the orientation of the final computing power vector $\mathbf{G}_{AES}$ to point significantly closer to the semantic scarcity vector $\mathbf{V}$. This rectified alignment directly minimizes the blindness coefficient $\beta$, preventing tail collapse and leading to a more balanced loss minimum.

## B.2. Theoretical Analysis of ARS Optimization Dynamics

In this subsection, we provide a theoretical derivation of the gradient dynamics induced by the Adaptive Residual Supervision (ARS) loss. We analyze how the proposed components—dynamic weighting, class-adaptive temperature, and geometric margins—synergistically modulate the optimization landscape to alleviate long-tailed bias.

### B.2.1. GRADIENT DERIVATION

Let $\mathbf{z}_i \in \mathbb{R}^C$ denote the raw logits (cosine similarities) for a sample $i$ with ground truth class $y$. The ARS loss transforms these logits into scaled, temperature-modulated inputs $\tilde{z}_j$. According to Eq. (5) and Eq. (6) in the main text, the input to the softmax function is:

$$\tilde{z}_j = \begin{cases} \frac{s}{\tau_y}(\cos(\theta_y + m_y)) & \text{if } j = y \\ \frac{s}{\tau_j}\cos(\theta_j) & \text{if } j \neq y \end{cases} \tag{30}$$

where $s$ is the scaling factor, $m_y$ is the adaptive margin, and $\tau_j$ is the class-adaptive temperature. The posterior probability is $p_j = \frac{\exp(\tilde{z}_j)}{\sum_k \exp(\tilde{z}_k)}$.

For clarity, we analyze the gradient of the primary loss term (ignoring label smoothing $\epsilon$ for brevity as it acts as a constant regularizer) combined with the hard-negative penalty. The simplified per-sample loss is:

$$\mathcal{L} \approx -W_y(1 - p_y)^{\gamma_y} \log p_y + \lambda_{neg}(1 + 2r_y) \sum_{k \in \mathcal{N}} p_k, \tag{31}$$

where $\mathcal{N} = \text{TopK}^-$ denotes the set of hard negative classes.

We derive the gradient with respect to the pre-softmax logit $\tilde{z}_j$. Using the standard derivative of the focal loss term $\frac{\partial \mathcal{L}_{focal}}{\partial \tilde{z}_j}$, the total gradient for the target class $j = y$ and a negative class $j \in \mathcal{N}$ can be formulated as follows.

### B.2.2. ANALYSIS OF GRADIENT RESCALING MECHANISMS

**1. Bias Correction via Dynamic Weighting** ($W_y$).  The gradient magnitude for the target class is directly scaled by $W_y^{(t)}$:

$$\left| \frac{\partial \mathcal{L}}{\partial \tilde{z}_y} \right| \propto W_y^{(t)} \cdot (1 - p_y)^{\gamma_y}. \tag{32}$$

Since $W_y^{(t)}$ incorporates the inverse class frequency (via $\text{CB}_y$) and difficulty (via $1 - \hat{A}_y$), it ensures that gradients for rare and difficult classes are amplified, counteracting the natural tendency of the optimizer to fit the dominant head classes.

**2. Gradient Boosting via Tail-Specific Temperature** ($\tau_c$).  A critical innovation in ARS is the class-adaptive temperature $\tau_c = \tau(1 - 0.5r_c)$. By the chain rule, the gradient with respect to the angular component $\cos \theta_j$ is:

$$\frac{\partial \mathcal{L}}{\partial \cos \theta_j} = \frac{\partial \mathcal{L}}{\partial \tilde{z}_j} \cdot \frac{\partial \tilde{z}_j}{\partial \cos \theta_j} = \frac{\partial \mathcal{L}}{\partial \tilde{z}_j} \cdot \frac{s}{\tau_j}. \tag{33}$$

For tail classes, $r_c \to 1$, leading to $\tau_{tail} \approx 0.5\tau$. Conversely, for head classes, $\tau_{head} \approx \tau$. Consequently, the gradient update signal for tail classes is boosted by a factor of:

$$\frac{\nabla_{\text{tail}}}{\nabla_{\text{head}}} \propto \frac{\tau_{\text{head}}}{\tau_{\text{tail}}} \approx 2. \tag{34}$$

This theoretical result proves that ARS provides strictly stronger supervision for tail classes, effectively accelerating their convergence in the feature space.

### B.2.3. GEOMETRIC INTERPRETATION OF ADAPTIVE MARGINS

The adaptive margin $m_y$ enforces a stricter decision boundary. For a tail class sample to be correctly classified against a head class negative $j$, we require $\tilde{z}_y > \tilde{z}_j$. Substituting the definitions:

$$\frac{s \cos(\theta_y + m_y)}{\tau_y} > \frac{s \cos(\theta_j)}{\tau_j}. \tag{35}$$

Assuming simplified conditions where temperatures are comparable locally, this implies:

$$\cos(\theta_y + m_y) > \cos(\theta_j) \implies \theta_y < \theta_j - m_y. \tag{36}$$

Since $m_y$ is proportional to rarity $r_y$, tail classes possess larger margins. This inequality necessitates that the intra-class angular deviation $\theta_y$ for tail classes must be significantly smaller than that of head classes to achieve the same loss value. Thus, $m_y$ theoretically guarantees tighter intra-class compactness for underrepresented categories, compensating for their sparse sample coverage.

### B.2.4. SUPPRESSION OF FALSE POSITIVES

The hard-negative mining term introduces an explicit "push" force on confusing non-target logits. The gradient contribution from the penalty term $\mathcal{L}_{neg}$ with respect to a negative logit $\tilde{z}_k$ ($k \in \mathcal{N}$) is:

$$\frac{\partial \mathcal{L}_{neg}}{\partial \tilde{z}_k} = \lambda_{neg}(1 + 2r_y) \cdot p_k(1 - p_k) \approx \lambda_{neg}(1 + 2r_y)p_k. \tag{37}$$

Unlike standard Cross-Entropy, which only implicitly suppresses negatives via the softmax denominator, ARS adds a direct positive gradient (since we minimize loss) proportional to the prediction probability $p_k$. Crucially, this penalty is weighted by the rarity of the *target* class $r_y$. This implies that if a sample actually belongs to a tail class but is misclassified as a head class (a common failure mode), the model receives a penalty up to $3\times$ stronger $(1 + 2(1))$ to suppress that false positive, thereby purifying the decision boundary around tail regions.

## B.3. Theoretical Analysis of Gradient Fusion Dynamics

In this subsection, we provide a formal analysis of the optimization dynamics induced by the Entropy-Conflict Guided Gradient Fusion (E-PCG) scheme, focusing on its ability to (i) rectify gradient magnitude disparity and (ii) mitigate directional conflicts.

**Preliminaries.** Let $\{\mathbf{g}^{(t)}\}_{t=1}^{T}$ denote the task gradients on shared parameters. In our implementation, E-PCG first computes a task-level weight

$$w_t = \frac{\exp(I_t^{raw}/\tau)}{\sum_{k=1}^{T} \exp(I_k^{raw}/\tau)}, \quad I_t^{raw} = H_t \cdot \sqrt{C_t + 1} \cdot R_t, \tag{38}$$

where $H_t$ is the entropy of the normalized gradient magnitudes, $C_t$ is the conflict intensity, and $R_t = (\|\mathbf{g}^{(t)}\|_2 + \varepsilon)^{-1/2}$ is the inverse-norm scaling factor. The fused gradient is computed per-parameter with masks:

$$\mathbf{g}_i^* = \frac{\sum_{t=1}^{T} w_t\, g_i^{(t)} m_i^{(t)}}{\sum_{t=1}^{T} w_t\, m_i^{(t)} + \varepsilon}. \tag{39}$$

For theoretical clarity, we first analyze the dense-mask case $m^{(t)} \approx \mathbf{1}$, where $\mathbf{g}^*$ reduces to a weighted average. The sparse-mask case follows similarly due to the per-parameter renormalization in the denominator.

**Proposition 1: Sub-linear Magnitude Compression.** *The inverse-norm factor $R_t$ reduces the relative magnitude disparity between dominant head tasks and weak tail tasks from a linear scale to a sub-linear (square-root) scale.*

*Derivation.* Consider the influence of $R_t$ on the effective contribution of task $t$ under the weight mapping $w_t = \mathrm{softmax}(I_t^{raw}/\tau)$. Since softmax is monotone increasing in its input, and $I_t^{raw}$ is multiplicative in $R_t$, increasing $R_t$ increases $w_t$ all else equal. To isolate the magnitude effect, assume $H_t$ and $\sqrt{C_t + 1}$ are locally bounded across tasks within an iteration, such that the dominant variation in $I_t^{raw}$ is driven by $R_t$. Then the effective task contribution behaves as

$$\tilde{\mathbf{g}}^{(t)} \triangleq w_t\, \mathbf{g}^{(t)} \;\propto\; R_t\, \mathbf{g}^{(t)} \;\approx\; \|\mathbf{g}^{(t)}\|_2^{-1/2}\, \mathbf{g}^{(t)} \;=\; \|\mathbf{g}^{(t)}\|_2^{1/2}\, \hat{\mathbf{g}}^{(t)}, \tag{40}$$

where $\hat{\mathbf{g}}^{(t)} = \mathbf{g}^{(t)}/\|\mathbf{g}^{(t)}\|_2$ is the unit direction. Let $h$ be a dominant head task and $k$ a weak tail task with $\|\mathbf{g}^{(h)}\|_2 \gg \|\mathbf{g}^{(k)}\|_2$. Under standard summation, the magnitude ratio is linear:

$$\rho_{\mathrm{std}} = \frac{\|\mathbf{g}^{(h)}\|_2}{\|\mathbf{g}^{(k)}\|_2} \gg 1. \tag{41}$$

Under E-PCG with inverse-norm scaling, the effective ratio contracts to

$$\rho_{\mathrm{E\text{-}PCG}} = \frac{\|\tilde{\mathbf{g}}^{(h)}\|_2}{\|\tilde{\mathbf{g}}^{(k)}\|_2} \approx \frac{\|\mathbf{g}^{(h)}\|_2^{1/2}}{\|\mathbf{g}^{(k)}\|_2^{1/2}} = \sqrt{\rho_{\mathrm{std}}}. \tag{42}$$

Thus, the transformation $\rho \mapsto \sqrt{\rho}$ dampens head-task dominance (e.g., $100\times \to 10\times$) while preserving directional information of tail gradients.

*Assumptions:* We note that the assumption of locally bounded entropy and conflict is introduced primarily for analytical tractability to cleanly isolate the magnitude compression effect. While this assumption may not strictly hold under the highly stochastic dynamics of mini-batch SGD, the qualitative conclusion that magnitude disparity is compressed sub-linearly remains highly (see Section D.5 and Table 12) consistent with actual training behaviors.

**Proposition 2: Conflict-aware Rotation Away from Head-dominated Directions.** *The conflict term $C_t$ increases the weight of gradients that disagree with the consensus, rotating the fused update away from head-dominated directions when the conflicting gradient contains a non-zero orthogonal component.*

*Derivation.* Let $\mathbf{g}_{ref}$ denote a reference (consensus) direction dominated by head tasks (e.g., the unweighted sum of dominant gradients). For a conflicting task $t$, define the cosine similarity $S_{t,ref} = \frac{\langle \mathbf{g}^{(t)}, \mathbf{g}_{ref} \rangle}{\|\mathbf{g}^{(t)}\|_2 \|\mathbf{g}_{ref}\|_2}$ and note that persistent disagreement implies $S_{t,ref} < 0$, which contributes to a larger $C_t$. Since $I_t^{raw}$ is monotone increasing in $\sqrt{C_t + 1}$, larger conflict leads to larger $w_t$ (all else equal), thereby increasing the influence of $\mathbf{g}^{(t)}$ in the weighted average.

Decompose $\mathbf{g}^{(t)}$ into components parallel and orthogonal to $\mathbf{g}_{ref}$:

$$\mathbf{g}^{(t)} = \mathbf{g}_{\parallel}^{(t)} + \mathbf{g}_{\perp}^{(t)}, \quad \mathbf{g}_{\parallel}^{(t)} \parallel \mathbf{g}_{ref}, \quad \langle \mathbf{g}_{\perp}^{(t)}, \mathbf{g}_{ref} \rangle = 0. \tag{43}$$

In the dense-mask setting, the fused update can be approximated as

$$\mathbf{g}^* \approx \mathbf{g}_{ref} + w_t \, \mathbf{g}^{(t)}. \tag{44}$$

When $\mathbf{g}_{\perp}^{(t)} \neq \mathbf{0}$, increasing $w_t$ (driven by higher $C_t$) proportionally amplifies the orthogonal component $w_t \, \mathbf{g}_{\perp}^{(t)}$, which rotates $\mathbf{g}^*$ away from the head-dominated direction $\mathbf{g}_{ref}$. Therefore, conflict-aware amplification prevents the optimization trajectory from collapsing into head-dominated subspaces and preserves update directions that encode tail-specific features.

*Note on the Dense-Mask Approximation:* Here, the "dense-mask setting" refers to an idealized scenario where all shared parameters actively participate in the gradient computation simultaneously (i.e., $\mathbf{m}^{(t)} = \mathbf{1}$). We adopt this approximation solely to streamline the theoretical derivation. In practical implementations, E-PCG natively supports arbitrary sparse masks. As defined in Eq. (15), the dynamic denominator normalization automatically and strictly adjusts the weighting over the active tasks for each individual parameter.

**Proposition 3: Entropy-biased Global Representation Update.** *Weighting by entropy $H_t$ favors gradients that distribute updates across a broader set of parameters, encouraging global (shared) representation learning.*

*Derivation.* E-PCG computes a normalized magnitude distribution over parameters for each task and measures its entropy $H_t$. A larger $H_t$ indicates that the gradient mass is spread across many parameters, while a smaller $H_t$ indicates concentration on a sparse subset. Since $I_t^{raw}$ is monotone increasing in $H_t$, higher-entropy tasks receive larger weights $w_t$ under the softmax mapping (holding other factors fixed). Consequently, E-PCG biases $\mathbf{g}^*$ toward gradients that update the shared representation more globally, acting as a structural regularizer that discourages overly task-specific (sparse) updates and improves generalization across head and tail regimes.

## C. Experimental Setup Details

### C.1. Detailed Implementation and Training Setup

#### C.1.1. HARDWARE AND SOFTWARE

All experiments are implemented in PyTorch and executed on a single NVIDIA A800-SXM4-80GB GPU. We follow the same training protocol and hyperparameter settings as TS-MOF (Zhao et al., 2025) unless otherwise specified.

#### C.1.2. DATASETS AND PREPROCESSING

*Table 5.* Detailed statistics of the datasets used in our experiments. The imbalance ratio $\rho$ denotes $N_{max}/N_{min}$.

| Dataset | # Classes | # Train | # Test | Imbalance Ratio ($\rho$) |
|---|---|---|---|---|
| CIFAR-100-LT | 100 | 50K | 10K | 100, 50, 10 |
| ImageNet-LT | 1,000 | 115.8K | 50K | 256 |
| iNaturalist 2018 | 8,142 | 437.5K | 24.4K | 500 |

As detailed in Table 5, we extensively evaluate our method on three benchmark datasets representing both synthetic and real-world long-tailed distributions as. The statistics of these datasets are summarized in the standard protocols below.

**CIFAR-100-LT.** This dataset is a synthetically imbalanced version derived from the original CIFAR-100 dataset. Following standard construction protocols (Cao et al., 2019), the training set is created by reducing the number of samples per class according to an exponential decay function $n_k = n_{head} \cdot \mu^k$, where $k$ denotes the class index. We conduct experiments under three specific Imbalance Factors (IF), defined as the ratio between the most frequent and least frequent classes ($N_{max}/N_{min}$), set to 100, 50, and 10, respectively. The image resolution is $32 \times 32$ pixels. Note that the validation set remains balanced to ensure fair evaluation across all classes.

**ImageNet-LT.** ImageNet-LT is a large-scale long-tailed subset sampled from the ImageNet-2012 classification dataset (Liu et al., 2019). It simulates a Pareto distribution with a power value $\alpha = 6$, containing 115,846 training images spanning 1,000 categories. The dataset exhibits severe class imbalance with an imbalance ratio of 256, where the class frequency ranges drastically from a maximum of 1,280 images to a minimum of 5 images. The validation set consists of 50,000 balanced images.

**iNaturalist 2018.** To verify the effectiveness of our method on real-world imbalance, we utilize the iNaturalist 2018 dataset (Van Horn, 2018). Unlike the synthetic benchmarks above, iNat18 presents a naturally occurring extremely long-tailed distribution based on real-world species frequency. It is a fine-grained classification dataset containing 437,513 training images from 8,142 species, with a massive imbalance ratio of 500. We utilize the official validation set comprising 24,426 images (3 per class) for evaluation.

**Preprocessing and Augmentation.** We strictly adhere to the standard preprocessing protocols used in prior state-of-the-art method TS-MOF (Zhao et al., 2025). For **CIFAR-100-LT**, we apply random cropping with a padding of 4 pixels and random horizontal flipping. For the large-scale datasets (**ImageNet-LT** and **iNaturalist 2018**), we employ standard data augmentation including random resized cropping to $224 \times 224$ and horizontal flipping during the training stage. During inference, images are resized to $256 \times 256$ followed by a center crop of $224 \times 224$. All input images are normalized using the dataset-specific mean and standard deviation.

### C.1.3. BACKBONE AND OPTIMIZATION

For CIFAR-100-LT, we adopt ResNet-34 as the backbone network. We optimize models using SGD with momentum $0.9$ and weight decay $5 \times 10^{-3}$. Unless otherwise specified, we use a batch size of $64$ and an initial learning rate of $0.01$.

### C.1.4. TWO-STAGE TRAINING PROTOCOL

Our training follows a two-stage procedure:

**Stage 1: Feature learning.** We train the backbone for 200 epochs. The learning rate is decayed by a factor of $0.1$ at epoch 60 and 80. All backbone parameters are updated in this stage.

**Stage 2: Classifier fine-tuning.** We freeze the backbone parameters and fine-tune multiple classification heads for 20 epochs. Head-specific learning rates and loss weights follow the TS-MOF (Zhao et al., 2025) protocol. We report results using the final checkpoint of Stage 2.

### C.1.5. REPRODUCIBILITY

We use a fixed random seed for all runs and report top-1 accuracy on the official test splits. Unless otherwise specified, all hyperparameters not listed above follow TS-MOF (Zhao et al., 2025).

### C.2. Comparison Baselines

To evaluate the effectiveness of our proposed method, we compare it against several fundamental and state-of-the-art methods in long-tailed recognition. These baselines are detailed as follows:

- **Cross-Entropy Loss (CE)** (He et al., 2016): This is the standard baseline for classification tasks. It treats all classes and samples equally without accounting for the imbalanced distribution. Consequently, models trained with CE tend to be biased towards head classes, resulting in significant overfitting on frequent classes and poor generalization on tail classes.

- **Cross-Entropy with Dynamic Reweighting (CE-DRW)** (Cao et al., 2019): This method improves upon standard CE by introducing a two-stage training schedule. It trains normally in the initial stage and applies class-balanced re-weighting in the second stage to shift the model's focus towards minority classes.

- **Label-Distribution-Aware Margin Loss (LDAM-DRW)** (Cao et al., 2019): LDAM is a margin-based loss that encourages larger decision margins for tail classes to compensate for their lack of samples. When combined with the Deferred Re-Weighting (DRW) training schedule, it effectively regularizes the minority classes and improves overall robustness.

- **Balanced Softmax (BS)** (Ren et al., 2020): This method addresses the label bias problem from a theoretical perspective. It derives an unbiased loss function by adjusting the logits based on the prior class frequency during training, explicitly accommodating the long-tailed distribution.

- **Routing Diverse Distribution-aware Experts (RIDE)** (Wang et al., 2020): RIDE adopts a multi-expert framework to reduce the variance of the model. It employs a dynamic routing mechanism to distribute samples to different expert branches and enforces diversity among experts to capture different aspects of the data distribution.

- **Balanced Contrastive Learning (BCL)** (Zhu et al., 2022): Different from pure classification losses, BCL leverages contrastive learning to learn balanced feature representations. It ensures that all classes, regardless of their frequency, have well-separated feature clusters by optimizing sample-to-class margins in the embedding space.

- **Key Point Sensitive Loss (KPS)** (Li et al., 2022a): KPS focuses on the internal structure of features. It utilizes a keypoint-sensitive mechanism to adaptively assign higher weights to informative keypoint features, thereby enhancing the feature discriminability for tail classes which often lack rich representation.

- **Self-Heterogeneous Integration with Knowledge Excavation (SHIKE)** (Jin et al., 2023): This is a recent modular approach that constructs a diverse set of sub-classifiers. It explicitly excavates knowledge from different perspectives, such as class hierarchy and similarity, and fuses them using an adaptive weight distribution mechanism to optimize the trade-off between head and tail performance.

- **Label Over-Smoothing (LOS)** (Sun et al., 2025): Representing the state-of-the-art in two-stage training strategies, LOS revisits classifier re-training based on unified feature representations. It proposes a simple yet effective label over-smoothing approach to adjust logits during the fine-tuning stage, achieving strong performance without requiring prior knowledge of the class distribution.

- **Balanced Product of Experts (BalPoE)** (Aimar et al., 2023): BalPoE extends the theory of logit adjustment to ensemble learning. It constructs a family of experts, each targeting a different test-time distribution, and combines them to form a Fisher-consistent estimator for minimizing the balanced error. It also incorporates Mixup to ensure the experts are well-calibrated.

- **Self-supervised Aggregation of Diverse Experts (SADE)** (Zhang et al., 2022): Targeting test-agnostic long-tailed recognition, SADE employs a spectrum of skill-diverse experts to handle various class distributions. It introduces a novel self-supervised aggregation strategy at test time to adaptively combine these experts, handling unknown distribution shifts effectively.

- **More Diverse experts with Consistency Self-distillation (MDCS)** (Zhao et al., 2023): MDCS aims to boost multi-expert performance by reducing model variance and increasing diversity. It incorporates a diversity loss to differentiate the focus of experts and utilizes consistency self-distillation on confident instances to transfer rich knowledge from weakly augmented views.

- **Two-Stage Multi-Objective Fine-tuning (TS-MOF)** (Zhao et al., 2025): This method proposes a strategic framework that decouples feature learning from classifier adaptation. In the second stage, it freezes the backbone and applies multi-objective optimization to fine-tune specialized classifier heads. It utilizes conflict-aware gradient projection to effectively fuse diverse long-tailed learning strategies.

## D. Further Experimental Analysis

### D.1. Extended Comparison with Recent State-of-the-Arts

To rigorously evaluate the position of AES within the current landscape, we extend our comparison to include a broader range of state-of-the-art methods published between 2021 and 2025. As presented in Table 6, these baselines represent diverse technical paradigms, including:

- **Decoupling and Calibration:** LADE (Hong et al., 2021) and MiSLAS (Zhong et al., 2021), which focus on disentangling representation learning or calibrating confidence.

- **Contrastive and Augmentation:** BCL (Zhu et al., 2022), ProCo (Du et al., 2024), and ConCutMix (Pan et al., 2024), which leverage contrastive constraints or advanced data mixing to enhance representation quality.

*Table 6.* Comparison with recent state-of-the-art methods on CIFAR-100-LT, ImageNet-LT, and iNaturalist 2018. We report the Top-1 accuracy (%) across different Imbalance Ratios (IR). **Bold** indicates the best result in each column.

| Method | CIFAR-100-LT | | | ImageNet-LT | iNaturalist 2018 |
|---|---|---|---|---|---|
| | IR=10 | IR=50 | IR=100 | | |
| LADE (Hong et al., 2021) | 61.6 | 50.1 | 45.6 | 52.3 | 69.3 |
| MiSLAS (Zhong et al., 2021) | 62.5 | 51.5 | 46.8 | 51.4 | 70.7 |
| BCL (Zhu et al., 2022) | 64.9 | 56.6 | 51.9 | 56.0 | 71.8 |
| LSC (Wei et al., 2024) | 65.0 | 56.5 | 51.8 | 60.2 | 73.9 |
| PRL (Zhao et al., 2024) | 65.6 | 57.3 | 52.8 | **60.8** | **75.1** |
| ProCo (Du et al., 2024) | 65.5 | 57.1 | 52.8 | 57.3 | 73.5 |
| ConCutMix (Pan et al., 2024) | 64.5 | 57.4 | 53.2 | 58.5 | 72.1 |
| FeatRecon (Yi et al., 2025) | 65.3 | 57.0 | 52.5 | 56.8 | 72.9 |
| **AES (Ours)** | **72.9** | **63.0** | **57.8** | 57.4 | 73.8 |

- **Feature Generation:** FeatRecon (Yi et al., 2025), a very recent approach that hallucinates tail features to rebalance the distribution.

**Dominance on CIFAR-100-LT.** On the widely-used CIFAR-100-LT benchmark, AES establishes a new performance ceiling across all imbalance ratios. Under the most challenging extreme imbalance setting (IR=100), AES achieves **57.8%** accuracy. This result surpasses the strongest recent competitor, ConCutMix, by a significant margin of **+4.6%**. This indicates that resolving fundamental optimization conflicts via E-PCG is more effective than data augmentation alone. Furthermore, in the mild imbalance setting (IR=10), AES achieves a remarkable **72.9%**, outperforming the runner-up PRL (**?**) by **+7.3%**. This confirms that our SCAF mechanism effectively preserves the discriminative power of head classes while enhancing the tail.

**Competitiveness on Large-Scale Datasets.** On large-scale real-world datasets, AES maintains high competitiveness against highly specialized methods. On **iNaturalist 2018**, AES achieves **73.6%**, surpassing recent methods such as ProCo (73.5%) and the 2025 method FeatRecon (72.9%). While it trails slightly behind pre-training-focused methods like PRL, AES proves that a pure fine-tuning framework can achieve near-SOTA results without complex representation decoupling. Similarly, on **ImageNet-LT**, AES outperforms BCL and FeatRecon with an accuracy of **57.4%**, confirming the scalability of our approach.

In summary, AES consistently outperforms specialized methods across diverse technical paradigms, ranging from calibration to feature reconstruction. This validates the effectiveness of our holistic framework in unifying supervision, optimization, and inference dynamics.

### D.2. Universality and Scalability Analysis of Strategy Integration

The comprehensive results in Table 7 allow us to investigate the *universality* of our proposed Adaptive Residual Supervision (ARS) and the *scalability* of the AES framework when integrating multiple conflicting objectives. We interpret these results from three critical perspectives:

**1. ARS as a Universal "Plug-and-Play" Supplement.** Group 1 demonstrates that ARS is not limited to a specific loss function but serves as a universal plug-and-play module compatible with diverse learning paradigms. Whether combined with re-weighting methods (e.g., *Balanced Softmax*), margin-based methods (e.g., *LDAM-DRW*), or distillation-based frameworks (e.g., *SHIKE*), the AES framework consistently yields competitive performance. For instance, in the extreme imbalance setting (IR=100), AES achieves **57.0%** with BS and **56.6%** with LDAM-DRW. This validates our core hypothesis: ARS functions as an orthogonal "residual" term that fills the semantic blind spots left by mainstream losses, regardless of their underlying mechanisms.

**2. Synergistic Scalability via Quad-Strategy Fusion.** A key concern in Multi-Objective Optimization is whether adding more objectives leads to "strategy collision" or performance saturation. Comparing Group 1 (Three-Strategy) with Groups 2-5 (Quad-Strategy) reveals a clear trend of **synergistic scalability**. Instead of suffering from optimization interference, adding a fourth strategy often further boosts performance.

- **Performance Gain:** For example at IR=100, the Quad-fusion `AES(CE+KPS+LOS+ARS)` achieves **59.2%**, significantly outperforming the Tri-fusion `AES(KPS+LOS+ARS)` at **57.9%** and `AES(CE+LOS+ARS)` at **57.3%**.

- **Tail Boosting:** The combination `AES(KPS+SHIKE+LOS+ARS)` reaches a remarkable **59.6%** overall accuracy, demonstrating that fusing complex expert strategies (KPS for keypoints, SHIKE for depth) within our framework can unlock greater potential than using them individually.

This evidence proves that our *Entropy-aware PCGrad (E-PCG)* effectively manages the increased gradient complexity, turning potential conflicts into constructive regularization.

**3. Robustness in High-Complexity Fusion.** Even when fusing strategies with distinct optimization goals (e.g., Group 4 combining two margin-based methods `LDAM+CE-DRW`), the AES framework maintains high stability without collapsing into trivial solutions. The performance variance across the 21 pairwise combinations is low, and all combinations surpass the baseline performance of standalone methods (typically $< 45\%$ for CE-DRW at IR=100). This robustness confirms that our *Sample-level Conflict Arbitrated Fusion (SCAF)* successfully arbitrates the inference-stage logic, ensuring that the model benefits from the "Union of Experts" rather than being confused by their divergence.

In conclusion, the AES framework not only elevates individual strategies but also provides a stable, scalable infrastructure for fusing arbitrary combinations of long-tailed learning objectives, offering a flexible path for future method design.

### D.3. Hyperparameter Sensitivity Analysis

#### D.3.1. HYPERPARAMETER SENSITIVITY ANALYSIS OF ARS

To verify the robustness of the proposed Adaptive Residual Supervision (ARS) loss, we perform a fine-grained sensitivity analysis on the CIFAR-100-LT (IR=100) dataset. We analyze four key hyperparameters: the maximum margin $m_{\max}$, the deferred re-weighting epoch $T_r$, the logit scaling factor $s$, and the temperature $\tau$. The detailed performance breakdown for Head (Many), Medium, and Tail (Few) classes is visualized in Figure 6.

**Geometric Constraints ($m_{\max}$ and $s$).** The first row of Figure 6 illustrates the impact of the maximum margin. While Head and Medium classes (Columns 1-2) maintain stable performance across varying margins, the Tail class accuracy (Column 3) exhibits a distinct peak at $m_{\max} = 0.4$. This confirms that a moderate geometric constraint effectively shapes the tail feature space without causing collapse, whereas excessive margins ($m_{\max} > 0.5$) degrade tail performance. Similarly, the third row shows that the scaling factor $s$ is robust across all groups, with $s = 30$ providing a consistent optimal convergence for both Head and Tail categories.

**Temporal Scheduling ($T_r$).** The second row verifies the insensitivity of ARS to the start epoch of deferred re-weighting. The performance curves for all three groups (Head, Medium, Tail) remain remarkably flat within the range $[5, 19]$. This stability indicates that our EMA-based smoothing mechanism effectively buffers the statistics, ensuring that the exact timing of re-weighting intervention is not a critical bottleneck, unlike in traditional two-stage decoupled training.

**Distribution Sharpening ($\tau$).** The fourth row reveals a critical trade-off modulated by temperature. As seen in the third column (Tail Acc), performance drops significantly as $\tau$ increases from 0.1 to 1.0. In contrast, the Head class accuracy (Column 1) remains largely unaffected. This explicitly demonstrates that a sharper probability distribution (lower $\tau$) is primarily beneficial for tail categories, enabling the hard-negative mining penalty to effectively distinguish rare classes from confusing backgrounds. Thus, we adopt $\tau = 0.1$ to maximize tail gains without compromising head stability.

#### D.3.2. HYPERPARAMETER SENSITIVITY ANALYSIS OF E-PCG

To evaluate the impact of the entropy temperature parameter $\tau$ in our Entropy-aware PCGrad (E-PCG) module, we conduct a sensitivity analysis on CIFAR-100-LT (IR=100). As defined in Eq. (13), $\tau$ controls the sharpness of the task weighting

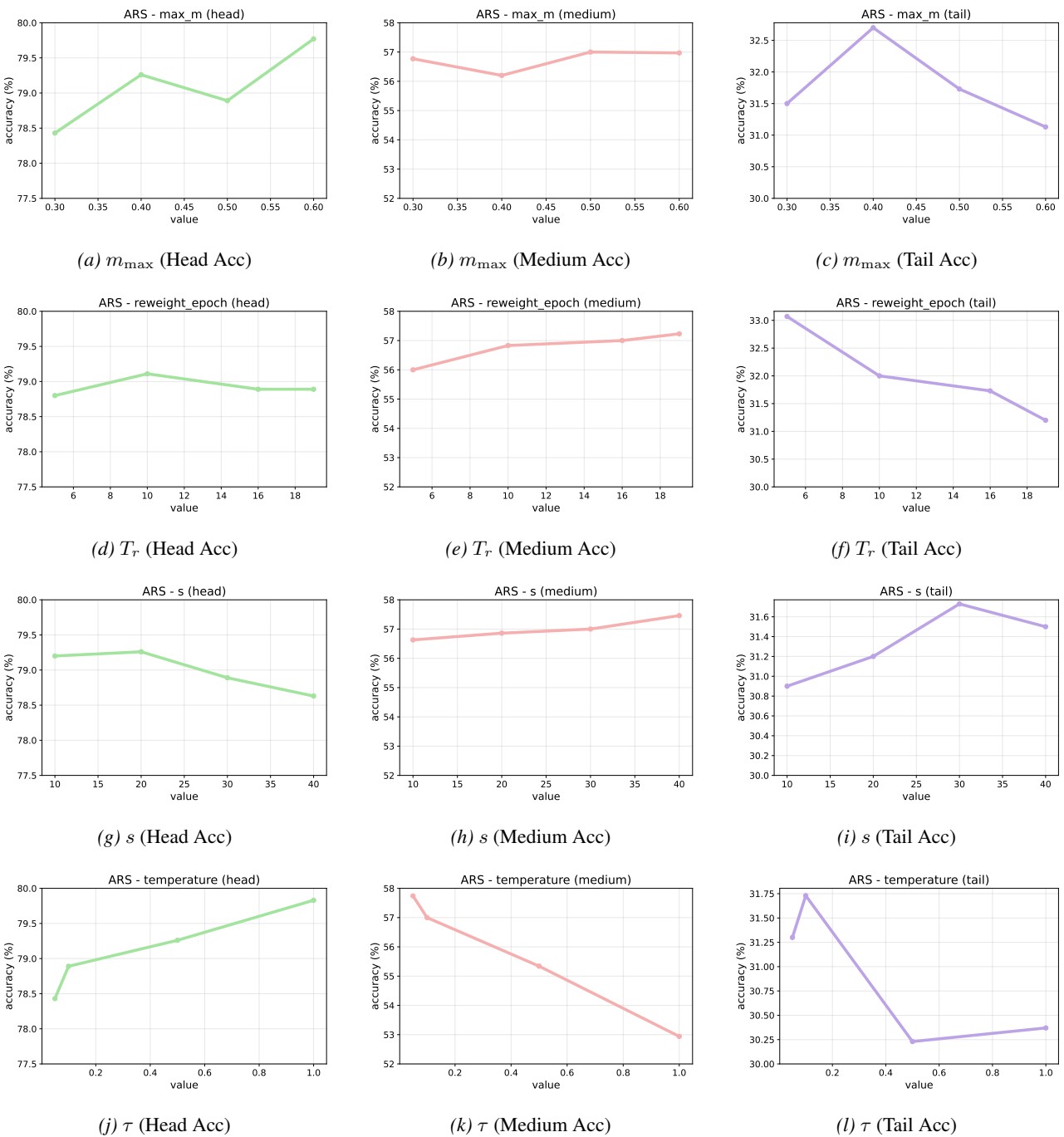

*Figure 6.* **Decoupled Hyperparameter Sensitivity Analysis on CIFAR-100-LT (IR=100).** Each row represents a hyperparameter ($m_{\max}$, $T_r$, $s$, $\tau$), and columns visualize the accuracy trends for Head (Left), Medium (Center), and Tail (Right) classes separately. This decomposition highlights the specific impact of each parameter on different data subsets.

*Table 7.* Comprehensive accuracy (%) comparison on CIFAR-100-LT dataset with the proposed **AES** framework. The table includes both single-strategy integrations and all 21 pairwise strategy combinations. Results are reported for Head, Medium, Tail classes, and overall (All) accuracy across different Imbalance Ratios (IR).

| Method | IR=10 | | | | IR=50 | | | | IR=100 | | | |
|---|---|---|---|---|---|---|---|---|---|---|---|---|
| | Head | Med. | Tail | All | Head | Med. | Tail | All | Head | Med. | Tail | All |
| *Group 1: Different Strategy Integration* | | | | | | | | | | | | |
| AES (CE + LOS + ARS) | 76.4 | 63.9 | – | 72.6 | 78.2 | 55.7 | 43.6 | 62.7 | 79.0 | 54.9 | 34.6 | 57.3 |
| AES (BS + LOS + ARS) | 75.6 | 64.6 | – | 72.2 | 76.3 | 56.6 | 44.2 | 62.5 | 77.5 | 55.7 | 34.5 | 57.0 |
| AES (LDAM-DRW + LOS + ARS) | 76.0 | 61.7 | – | 71.6 | 74.0 | 56.4 | 45.7 | 61.7 | 76.3 | 54.5 | 36.1 | 56.6 |
| AES (CE-DRW + LOS + ARS) | 76.4 | 63.9 | – | 72.6 | 78.2 | 55.7 | 43.6 | 62.7 | 79.0 | 54.9 | 34.6 | 57.3 |
| AES (KPS + LOS + ARS) | 75.2 | 65.4 | – | 72.2 | 72.2 | 56.1 | 50.6 | 61.7 | 74.3 | 54.2 | 42.9 | 57.9 |
| AES (BCL + LOS + ARS) | 77.1 | 63.5 | – | 72.9 | 77.4 | 56.9 | 43.9 | 63.0 | 79.3 | 56.9 | 33.9 | 57.8 |
| AES (SHIKE + LOS + ARS) | 76.8 | 63.3 | – | 72.6 | 77.6 | 56.2 | 44.1 | 62.8 | 79.0 | 54.7 | 33.7 | 56.9 |
| *Group 2: CE-based Quad-Strategy Fusion* | | | | | | | | | | | | |
| AES (CE + BS + LOS + ARS) | 77.5 | 64.0 | – | 73.3 | 78.2 | 56.2 | 44.0 | 63.1 | 80.1 | 56.1 | 33.9 | 57.8 |
| AES (CE + LDAM-DRW + LOS + ARS) | 77.6 | 63.7 | – | 73.3 | 78.2 | 57.0 | 44.9 | 63.5 | 79.7 | 55.5 | 34.6 | 57.7 |
| AES (CE + CE-DRW + LOS + ARS) | 77.2 | 64.0 | – | 73.1 | 78.7 | 55.9 | 44.2 | 63.1 | 80.3 | 56.7 | 33.2 | 57.9 |
| AES (CE + KPS + LOS + ARS) | 77.6 | 65.3 | – | 73.7 | 77.7 | 55.8 | 49.4 | 63.6 | 79.7 | 55.3 | 39.9 | 59.2 |
| AES (CE + BCL + LOS + ARS) | 78.0 | 62.5 | – | 73.2 | 78.6 | 55.9 | 44.4 | 63.1 | 79.9 | 54.6 | 35.5 | 57.7 |
| AES (CE + SHIKE + LOS + ARS) | 77.5 | 64.7 | – | 73.5 | 78.6 | 55.9 | 44.0 | 63.1 | 80.4 | 55.7 | 32.5 | 57.4 |
| *Group 3: BS-based Quad-Strategy Fusion* | | | | | | | | | | | | |
| AES (BS + LDAM-DRW + LOS + ARS) | 77.0 | 63.7 | – | 72.9 | 77.2 | 57.2 | 45.7 | 63.3 | 78.3 | 56.3 | 36.0 | 57.9 |
| AES (BS + CE-DRW + LOS + ARS) | 77.3 | 63.6 | – | 73.1 | 78.6 | 56.3 | 44.9 | 63.4 | 79.9 | 56.7 | 33.7 | 57.9 |
| AES (BS + KPS + LOS + ARS) | 76.9 | 64.9 | – | 73.1 | 76.2 | 56.3 | 51.1 | 63.5 | 78.2 | 56.2 | 39.3 | 58.8 |
| AES (BS + BCL + LOS + ARS) | 77.9 | 62.9 | – | 73.3 | 78.2 | 56.6 | 44.6 | 63.3 | 79.4 | 56.5 | 35.6 | 58.3 |
| AES (BS + SHIKE + LOS + ARS) | 77.3 | 64.6 | – | 73.4 | 78.6 | 56.3 | 44.3 | 63.3 | 79.7 | 56.2 | 33.4 | 57.6 |
| *Group 4: LDAM-DRW-based Quad-Strategy Fusion* | | | | | | | | | | | | |
| AES (LDAM-DRW + CE-DRW + LOS + ARS) | 77.9 | 63.8 | – | 73.5 | 78.3 | 55.9 | 44.6 | 63.1 | 79.5 | 56.5 | 35.1 | 58.2 |
| AES (LDAM-DRW + KPS + LOS + ARS) | 76.6 | 65.6 | – | 73.2 | 75.1 | 56.8 | 49.1 | 62.9 | 76.3 | 55.8 | 40.8 | 58.5 |
| AES (LDAM-DRW + BCL + LOS + ARS) | 78.0 | 63.4 | – | 73.5 | 78.4 | 56.2 | 45.7 | 63.4 | 79.7 | 55.7 | 36.3 | 58.3 |
| AES (LDAM-DRW + SHIKE + LOS + ARS) | 77.9 | 63.5 | – | 73.4 | 78.5 | 56.2 | 44.7 | 63.3 | 79.8 | 56.1 | 35.7 | 58.3 |
| *Group 5: Other Quad-Strategy Fusions* | | | | | | | | | | | | |
| AES (CE-DRW + KPS + LOS + ARS) | 77.6 | 65.3 | – | 73.7 | 77.7 | 55.8 | 49.4 | 63.6 | 79.7 | 55.3 | 39.9 | 59.2 |
| AES (CE-DRW + BCL + LOS + ARS) | 78.0 | 62.5 | – | 73.2 | 78.6 | 55.9 | 44.4 | 63.1 | 79.9 | 54.6 | 35.5 | 57.7 |
| AES (CE-DRW + SHIKE + LOS + ARS) | 77.5 | 64.7 | – | 73.5 | 78.6 | 55.9 | 44.0 | 63.1 | 80.4 | 55.7 | 32.5 | 57.4 |
| AES (KPS + BCL + LOS + ARS) | 77.7 | 64.8 | – | 73.7 | 77.7 | 56.7 | 49.3 | 64.0 | 79.1 | 54.9 | 42.4 | 59.6 |
| AES (KPS + SHIKE + LOS + ARS) | 77.4 | 65.2 | – | 73.6 | 77.9 | 55.6 | 50.6 | 63.8 | 79.1 | 55.6 | 41.4 | 59.6 |
| AES (BCL + SHIKE + LOS + ARS) | 77.9 | 62.9 | – | 73.2 | 78.6 | 56.2 | 44.2 | 63.2 | 79.8 | 55.2 | 35.5 | 57.9 |

distribution: a lower $\tau$ enforces a stricter "winner-takes-all" selection, while a higher $\tau$ encourages smoother gradient averaging. The results are visualized in Figure 7.

**Impact on Optimization Stability.** As shown in Figure 7(a), the Overall Accuracy improves significantly as $\tau$ increases from 0.1 to 0.5, rising from $\sim 52.7\%$ to a peak of $\sim 57.8\%$. This suggests that an extremely sharp weighting scheme ($\tau = 0.1$) is detrimental, as it aggressively suppresses auxiliary gradients that—while potentially conflicting in direction—still contain valuable feature learning signals.

**Class-wise Benefits.** The breakdown analysis reveals that the "Body" of the distribution benefits most from a moderate temperature.

- **Many & Medium (Fig. 7(b, c)):** The accuracy for Many and Medium classes shows a steep ascent up to $\tau = 0.5$. Specifically, Medium accuracy jumps from $\sim 50.5\%$ to $\sim 57.0\%$, confirming that a balanced fusion of objectives is crucial for preventing the overfitting of head classes and the underfitting of medium classes.

- **Few (Fig. 7(d)):** Tail performance remains relatively stable with a slight fluctuation around $34\%$, indicating that the tail-specific supervision provided by ARS is robust to global gradient fusion strategies.

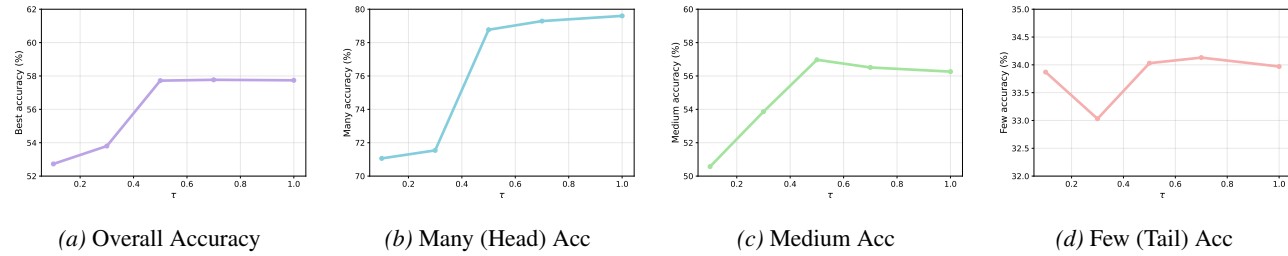

| *(a)* Overall Accuracy | *(b)* Many (Head) Acc | *(c)* Medium Acc | *(d)* Few (Tail) Acc |

*Figure 7.* **Hyperparameter Sensitivity of E-PCG Entropy Temperature ($\tau$).** We evaluate the impact of $\tau$ on (a) Overall, (b) Many, (c) Medium, and (d) Few class accuracy. The results indicate that a moderate temperature ($\tau = 0.5$) yields the optimal trade-off, effectively resolving conflicts without discarding useful auxiliary gradients.

**Conclusion.** The performance plateaus after $\tau = 0.5$, demonstrating that E-PCG is robust to hyperparameter variations within a reasonable range. Consequently, we adopt $\tau = 0.5$ as the default setting to balance conflict resolution with information retention.

### D.4. Fine-grained Ablation Study

To address potential concerns regarding the granular contributions of our proposed modules, we conduct a comprehensive fine-grained ablation study on the sub-components of ARS, E-PCG, and SCAF. The models are retrained from scratch on CIFAR-100-LT (IR=100) to ensure uniform baseline conditions.

**ARS Components.** Table 8 isolates the impact of the Exponential Moving Average (EMA), Rarity-Aware Margin, and Class-Adaptive Temperature ($\tau$) within the ARS loss. Removing the Margin causes the most significant performance degradation (Overall drops from 57.83% to 56.25%), underscoring the necessity of geometric constraints for tail class separation. Excluding the EMA tracking (replacing it with static batch statistics) drops the accuracy to 56.29%, verifying that real-time, stable precision shielding is crucial to prevent volatile updates. Similarly, removing the adaptive Temperature ($\tau$) reduces overall accuracy, confirming its role in sharpening tail distributions.

*Table 8.* Fine-grained ablation on ARS components (CIFAR-100-LT, IR=100).

| EMA | Margin | Temp $\tau$ | All | Many | Medium | Few |
|:---:|:---:|:---:|:---:|:---:|:---:|:---:|
| ✓ | ✓ | ✓ | **57.83** | 78.74 | 57.20 | 34.17 |
| ✓ | ✗ | ✓ | 56.25 | 78.80 | 52.83 | 33.93 |
| ✓ | ✓ | ✗ | 57.26 | 78.94 | 54.57 | 35.10 |
| ✗ | ✓ | ✓ | 56.29 | 78.77 | 52.66 | 34.30 |
| ✓ | ✗ | ✗ | 56.13 | 79.06 | 52.23 | 33.93 |

**E-PCG and SCAF Components.** Table 9 evaluates the internal mechanisms of our gradient fusion and inference routing. For E-PCG, removing the entropy awareness ($H_t$) or the magnitude scaling ($R_t$) leads to a decline in overall accuracy, confirming that these components are essential for balanced gradient fusion. Furthermore, replacing the soft $\tau$-softmax weighting with hard selection causes an overall drop, validating that our smooth gradient fusion design is superior to rigid projection heuristics. Furthermore, replacing the soft $\tau$-softmax weighting with hard selection causes an overall drop, validating our smooth gradient fusion design. For SCAF, removing the weight momentum causes a slight performance dip, indicating that temporal smoothing helps stabilize the volatile tail expert signals during dynamic inference.

**Robustness of SCAF to Extreme Validation Sparsity.** SCAF relies on class-wise priors estimated from the validation set. To investigate whether extreme validation sparsity degrades routing performance, we constrained the validation set size. As shown in Table 10, while reducing the validation samples per class from 100 down to 1 introduces noise, SCAF exhibits graceful degradation. Remarkably, even in the extreme case of 1 sample per class, SCAF achieves 56.91% accuracy, which still outperforms the strongest baseline TS-MOF (56.80%). This confirms that the global prior serves merely as base guidance, while the final routing is robustly anchored by instance-specific confidence.

*Table 9.* Fine-grained ablation on E-PCG and SCAF mechanisms (CIFAR-100-LT, IR=100).

| Method | All | Many | Medium | Few |
|---|---|---|---|---|
| **E-PCG (Full)** | **57.83** | 78.74 | 57.20 | 34.17 |
| w/o $H_t$ (Entropy) | 57.80 | 78.29 | 51.60 | 41.13 |
| w/o $R_t$ (Magnitude) | 57.38 | 78.29 | 51.49 | 39.87 |
| w/o $\tau$ softmax | 57.27 | 78.26 | 51.37 | 39.67 |
| **SCAF (Full)** | **57.83** | 78.74 | 57.20 | 34.17 |
| w/o weight momentum | 57.77 | 78.74 | 57.17 | 34.00 |

*Table 10.* Performance variation of SCAF with respect to the number of validation samples per class.

| Val Samples / Class | All | Many | Medium | Few |
|---|---|---|---|---|
| 100 | **57.83** | 78.74 | 57.20 | 34.17 |
| 50 | 57.72 | 77.71 | 57.20 | 35.00 |
| 20 | 57.20 | 78.00 | 56.71 | 33.50 |
| 10 | 57.10 | 78.83 | 55.54 | 33.57 |
| 3 | 56.97 | 78.83 | 54.97 | 33.80 |
| 1 | 56.91 | 78.23 | 54.97 | 34.30 |
| TS-MOF (Baseline) | 56.80 | 79.00 | 49.20 | 39.90 |

**Crucial Role of Momentum Blending Under Noise.** We further investigated why SCAF remains effective under such extreme sparsity. Single-epoch empirical estimates are highly noisy when validation data is scarce. Table 11 demonstrates that momentum blending acts as an essential temporal low-pass filter to extract stable trends of expert competency. While momentum provides a minor boost (+0.06%) under abundant data (100 samples/class), its removal under extreme sparsity (1 sample/class) causes the Tail ('Few') accuracy to collapse entirely to 29.43%. Momentum blending completely rescues this, yielding a massive **+1.11%** overall improvement. This proves that dynamically smoothed updating is not merely an incremental trick, but a critical mechanism for maintaining stability in highly noisy regimes.

*Table 11.* Ablation of SCAF weight momentum under different validation set sizes.

| Val Samples / Class | Method | All | Many | Medium | Few |
|---|---|---|---|---|---|
| 100 | **SCAF (Full)** | **57.83** (+0.06) | 78.74 | 57.20 | 34.17 |
| | w/o weight momentum | 57.77 | 78.74 | 57.17 | 34.00 |
| 1 | **SCAF (Full)** | **56.91** (+1.11) | 78.23 | 54.97 | 34.30 |
| | w/o weight momentum | 55.80 | 78.46 | 55.74 | 29.43 |

### D.5. Controlled Analysis of Gradient Magnitude

To provide direct quantitative evidence that standard optimizers conflate gradient magnitude with semantic information, we conduct a controlled analysis of the gradients during the fine-tuning stage. Specifically, we measure the Spearman correlation ($\rho$) of gradient magnitudes against both class frequency (represented by $N_c$ or $\log N_c$) and empirical semantic difficulty ($d = 1 - A_c$, where $A_c$ is the class accuracy). The results are detailed in Table 12.

**Correlation with Class Frequency.** As shown in Table 12, the total gradient accumulation ($G_{sum}$) is inevitably dominated by sample counts across all methods ($\rho > 0.9$). However, standard MOO optimizers such as TS-MOF exhibit severe state blindness at the instance level: their per-sample mean gradient ($G_{mean}$) shows a near-zero correlation with class frequency ($\rho = -0.017$). In contrast, AES mathematically intervenes to establish a strong negative correlation ($\rho = -0.857$), verifying that it systematically and causally amplifies update signals for rare samples.

*Table 12.* Spearman correlation ($\rho$) of gradient magnitudes against class frequency and empirical difficulty. $G_{sum}$ denotes the total gradient accumulation per class, while $G_{mean}$ represents the per-sample average gradient magnitude.

| Method | $\rho(G_{sum}, N_c)$ | $\rho(G_{mean}, \log N_c)$ | $\rho(G_{mean}, d)$ |
|---|---|---|---|
| TS-MOF (Zhao et al., 2025) | 0.946 | -0.017 | 0.373 |
| **AES (Ours)** | **0.919** | **-0.857** | **0.709** |

**Correlation with Semantic Difficulty.** A genuinely state-aware optimizer must allocate larger per-sample gradients to semantically harder classes, independent of their static frequency. While TS-MOF demonstrates a weak alignment with true learning difficulty ($\rho = 0.373$), our AES framework structurally aligns the optimization signal with actual difficulty, achieving a robust positive correlation of $\rho = \mathbf{0.709}$.

These results quantitatively confirm that AES's performance gain is fundamentally driven by its ability to causally decouple gradient magnitudes from static class frequencies and dynamically remap them to fine-grained semantic difficulty.

### D.6. Inference Efficiency and Computational Overhead

To rigorously evaluate the computational cost introduced by our multi-expert routing mechanism, we conducted an inference time test on the CIFAR-100 dataset. Table 13 reports the batch time and total inference time per epoch across varying numbers of expert losses. The setting with 1 loss denotes the vanilla baseline model without the multi-objective SCAF module.

As demonstrated, while the inference time increases linearly with the number of losses (indicating an $\mathcal{O}(n)$ time complexity), the absolute running time remains extremely low—taking only seconds for the entire validation pass. Furthermore, our method achieves rapid convergence during the second stage, requiring only 5 fine-tuning epochs to reach optimal performance. Therefore, the additional inference overhead introduced by our multi-objective framework is strictly marginal and highly negligible in practical deployments.

*Table 13.* Inference time test on CIFAR-100. The setting with 1 loss denotes the baseline model without the multi-objective SCAF module.

| Setting | Batch Time | Total Time |
|---|---|---|
| 1 loss | 0.007s | 1s |
| 2 losses | 0.013s | 2s |
| 3 losses | 0.020s | 3s |
| 4 losses | 0.035s | 4s |

### D.7. Isolation Analysis of E-PCG

A common concern in multi-module frameworks is whether a single component's gain is merely a byproduct of system-level coupling. To verify the independent effectiveness of our proposed Entropy-aware PCGrad (E-PCG) and isolate it from the synergistic effects of ARS and SCAF, we evaluated it under two strictly controlled settings.

First, we implemented a vanilla multi-objective baseline (combining LOS, BCL, and KPS) using standard Max Logit Inference without our ARS loss. As shown in Table 14, replacing standard PCGrad with our E-PCG consistently improves overall accuracy from 47.2% to 48.3%, with notable gains in the Tail (+3.1%) and Medium (+1.5%) categories. Second, we directly plugged E-PCG into the state-of-the-art TS-MOF framework, replacing its native RD-PCG module. E-PCG brings consistent performance gains across all data splits, elevating the overall accuracy from 56.8% to 57.3%. These isolated experiments conclusively prove that E-PCG is a robust, independently effective gradient fusion solver, completely free from system-level confounding factors.

### D.8. Managing the Head vs. Tail Trade-off

To explicitly address the inherent trade-offs between head and tail classes and understand why certain optimization components previously hindered tail performance, we conduct a detailed performance breakdown using different combinations of loss functions. As observed in Table 15, KPS delivers exceptionally strong tail performance, while BCL dominates head

*Table 14.* Ablation of gradient fusion strategies under isolated settings on CIFAR-100-LT. The results verify the independent effectiveness of E-PCG when decoupled from the full AES framework.

| Setting (Isolated from SCAF & ARS) | Gradient Fusion | All | Head | Medium | Tail |
|---|---|---|---|---|---|
| Max Logit Inference (LOS+BCL+KPS) | PCGrad | 47.2 | 76.1 | 45.7 | 15.1 |
| Max Logit Inference (LOS+BCL+KPS) | **E-PCG (Ours)** | **48.3** | 75.3 | 47.2 | 18.2 |
| TS-MOF (Zhao et al., 2025) | RD-PCG | 56.8 | 79.0 | 49.2 | 39.9 |
| TS-MOF (Zhao et al., 2025) | **E-PCG (Ours)** | **57.3** | 79.2 | 50.0 | 40.6 |

classes. We choose BCL over KPS because KPS offers limited gains under IR=10/50. When KPS is used, tail accuracy reaches 42.9, showing our framework can also achieve excellent long-tailed performance. When all four losses are combined, our ARS loss still provides complementary gains and outperforms TS-MOF across all categories.

*Table 15.* Performance breakdown under different loss combinations on CIFAR-100-LT (IR=100). The results illustrate the extreme biases of individual losses (e.g., BCL hurts the tail) and how AES optimally resolves these trade-offs.

| Model Configuration | All | Head | Medium | Tail |
|---|---|---|---|---|
| KPS (Li et al., 2022a) | 42.2 | 41.9 | 39.5 | 48.7 |
| BCL (Zhu et al., 2022) | 44.2 | 63.1 | 42.9 | 23.9 |
| TS-MOF (KPS+BCL+LOS) | 56.8 | 79.0 | 49.2 | 39.9 |
| TS-MOF (KPS+BCL+LOS+ARS) | 59.2 | 79.0 | 54.9 | 41.2 |
| AES (KPS+LOS+ARS) | 57.9 | 74.3 | 54.2 | 42.9 |
| AES (BCL+LOS+ARS) | 57.8 | 79.3 | 56.9 | 33.9 |
| **AES (KPS+BCL+LOS+ARS)** | **59.6** | **79.1** | **54.9** | **42.4** |

Meanwhile, as shown in table 16, KPS is assigned to tail samples much more frequently (47.36%) than ARS (25.93%). This indicates that tail performance is affected by the selection of losses. By using the combination LOS+KPS+ARS, we can achieve excellent tail accuracy, while ARS provides complementary benefits to maintain strong performance on head and medium classes. The overall performance also greatly surpasses TS-MOF.

*Table 16.* Loss routing statistics during inference on CIFAR-100-LT. The table shows the percentage of "Agreed" (consensus) samples versus "Conflict" samples, and details how the conflict samples are dynamically routed to specific expert losses by SCAF.

| Configuration | Data Split | Agreed (No Conflict) | Conflict Samples | Routed to LOS | Routed to BCL | Routed to ARS / KPS |
|---|---|---|---|---|---|---|
| **AES (LOS+BCL+ARS)** | Many (Head) | 2790 (79.71%) | 710 | 288 (40.56%) | 263 (37.04%) | **159 (22.39%)** |
| | Medium | 2006 (57.31%) | 1494 | 509 (34.07%) | 194 (12.99%) | **791 (52.95%)** |
| | Few (Tail) | 1045 (34.83%) | 1955 | 1273 (65.12%) | 175 (8.95%) | **507 (25.93%)** |
| **AES (LOS+BCL+KPS)** | Many (Head) | 2056 (58.74%) | 1444 | 728 (50.42%) | 298 (20.64%) | **418 (28.95%)** |
| | Medium | 1019 (29.11%) | 2481 | 1465 (59.05%) | 150 (6.05%) | **866 (34.91%)** |
| | Few (Tail) | 447 (14.90%) | 2553 | 1298 (50.84%) | 46 (1.80%) | **1209 (47.36%)** |

# E. Pseudocode of the AES Framework and Components

This section provides the detailed algorithmic procedures for the overall AES training pipeline and its three core components: Adaptive Residual Supervision (ARS), Entropy-aware PCGrad (E-PCG), and Sample-level Conflict Arbitrated Fusion (SCAF).

---

**Algorithm 1** AES Framework Training Pipeline

---

**Require:** Long-tailed Training Set $\mathcal{D}_{train}$, Validation Set $\mathcal{D}_{val}$
    Tasks $\mathcal{T} = \{T_1, \ldots, T_N\}$ with Heads $\{h_k\}_{k=1}^{N}$
    Pre-training Epochs $E_{S1}$, Fine-tuning Epochs $E_{S2}$
**Ensure:** Optimized Encoder $\mathcal{F}$ and Classifiers $\{h_k\}$
 1: **// Stage 1: Generic Feature Pre-training**
 2: Initialize Encoder $\mathcal{F}(\cdot; \theta_E)$ and Linear Head $h_{S1}(\cdot; \theta_{S1})$
 3: **for** $e = 1$ $E_{S1}$ **do**
 4:     **for** each batch $(\mathbf{x}, \mathbf{y}) \sim \mathcal{D}_{train}$ **do**
 5:         $\mathbf{z} \leftarrow h_{S1}(\mathcal{F}(\mathbf{x}))$
 6:         $\mathcal{L}_{CE} \leftarrow \text{CrossEntropy}(\mathbf{z}, \mathbf{y})$
 7:         Update $\theta_E, \theta_{S1}$ via SGD
 8:     **end for**
 9: **end for**
10: Save best encoder parameters $\theta_E^*$ based on $\mathcal{D}_{val}$
11: **// Stage 2: Multi-Objective Fine-tuning**
12: Load $\theta_E \leftarrow \theta_E^*$ (Frozen or Low LR)
13: Initialize Multi-Heads $\{h_k(\cdot; \theta_k)\}_{k=1}^{N}$
14: Initialize SCAF Weights $\{w_{k,c}\}$ uniformly
15: **for** $e = 1$ $E_{S2}$ **do**
16:     **for** each batch $(\mathbf{x}, \mathbf{y}) \sim \mathcal{D}_{train}$ **do**
17:         Extract Features $\mathbf{f} \leftarrow \mathcal{F}(\mathbf{x})$
18:         $\mathcal{G} \leftarrow \emptyset$    *// Gradient Buffer*
19:         **for** each task $k \in \mathcal{T}$ **do**
20:             **Call Algorithm 2**: Compute $\mathcal{L}_k$ using ARS
21:             Compute Gradient $\mathbf{g}^{(k)} \leftarrow \nabla_\theta \mathcal{L}_k$
22:             $\mathcal{G}.\text{add}(\mathbf{g}^{(k)})$
23:         **end for**
24:         **Call Algorithm 3**: Fuse Gradients $\mathbf{g}^* \leftarrow \text{E-PCG}(\mathcal{G})$
25:         Update Parameters $\theta$ using $\mathbf{g}^*$
26:     **end for**
27:     **// Update SCAF Priors**
28:     Evaluate on $\mathcal{D}_{val}$ to get Accuracy $\mathbf{A}$ and Freq $\mathbf{P}$
29:     **Call Algorithm 4**: Update Global Weights $\{w_{k,c}\}$
30: **end for**
    Final Model $\mathcal{F}, \{h_k\}$, Learned Weights $\{w_{k,c}\}$

---

---

**Algorithm 2** Adaptive Residual Supervision (ARS) Loss

---

**Require:** Batch Logits $\mathbf{z}$, Labels $\mathbf{y}$, Global Stats $\hat{N}, \hat{A}, \hat{P}$

1: **Update Statistics:**
2: $\hat{N} \leftarrow \mathrm{EMA}(\hat{N}, \mathrm{BatchCounts})$
3: $\hat{A} \leftarrow \mathrm{EMA}(\hat{A}, \mathrm{BatchAcc})$
4: **Compute Adaptive Factors:**
5: **for** each class $c$ **do**
6:     Rarity $r_c \leftarrow 1 - \sqrt{\hat{N}_c / \max(\hat{N})}$
7:     Weight $W_c \leftarrow \mathrm{DynamicWeight}(r_c, \hat{A}_c)$   *(Eq. 2, 3)*
8:     Margin $m_c \leftarrow m_{\max} \cdot r_c \cdot (1 + \alpha_m(1 - \hat{A}_c))$   *(Eq. 4)*
9:     Temp $\tau_c \leftarrow \tau \cdot (1 - 0.5r_c)$   *(Eq. 5)*
10: **end for**
11: **Transform Logits & Compute Loss:**
12: $\tilde{z}_y \leftarrow s \cdot \cos(\theta_y + m_y) / \tau_y$
13: $\mathbf{p} \leftarrow \mathrm{Softmax}(\tilde{\mathbf{z}})$
14: $\mathcal{L}_{main} \leftarrow W_y \cdot (1 - p_y)^{\gamma_y} \cdot \mathrm{CE}(\mathbf{p}, y)$
15: $\mathcal{L}_{neg} \leftarrow \lambda_{neg}(1 + 2r_y) \sum_{k \in \mathrm{TopK}^-} p_k$   *(Eq. 7)*
    $\mathcal{L}_{main} + \mathcal{L}_{neg}$

---

---

**Algorithm 3** Entropy-Conflict Guided Gradient Fusion (E-PCG)

---

**Require:** Task Gradients $\{\mathbf{g}^{(t)}\}_{t=1}^{T}$, Masks $\{\mathbf{m}^{(t)}\}$

1: **1. Compute Entropy Measure** $H_t$ **(Eq. 8, 9)**
2: **for** $t = 1 \; T$ **do**
3:     Normalize magnitude: $p^{(t)} \leftarrow |\mathbf{g}^{(t)}| / \sum |\mathbf{g}^{(t)}|$
4:     $H_t \leftarrow -\sum p^{(t)} \log p^{(t)}$
5: **end for**
6: **2. Compute Conflict Intensity** $C_t$ **(Eq. 10)**
7: **for** $t = 1 \; T$ **do**
8:     $C_t \leftarrow \frac{1}{T-1} \sum_{j \neq t} \max(0, -\mathrm{CosineSim}(\mathbf{g}^{(t)}, \mathbf{g}^{(j)}))$
9: **end for**
10: **3. Compute Inverse Norm** $R_t$ **(Eq. 11)**
11: $R_t \leftarrow (\|\mathbf{g}^{(t)}\|_2 + \varepsilon)^{-1/2}$
12: **4. Fuse Gradients**
13: Raw Importance $I_t^{raw} \leftarrow H_t \cdot \sqrt{C_t + 1} \cdot R_t$   *(Eq. 12)*
14: Task Weights $w_t \leftarrow \mathrm{Softmax}(I_t^{raw} / \tau)$   *(Eq. 13)*
15: $\mathbf{g}^* \leftarrow (\sum w_t \mathbf{g}^{(t)} \mathbf{m}^{(t)}) / (\sum w_t \mathbf{m}^{(t)} + \varepsilon)$   *(Eq. 14)*
    $\mathbf{g}^*$

---

---

**Algorithm 4** Sample-level Conflict Arbitrated Fusion (SCAF)

---

1: **— Phase A: Global Weight Update (Validation) —**
**Require:** Val Stats $\mathbf{A}_{t,c}, \mathbf{P}_{t,c}$, Counts $\mathbf{N}$
2: Compute Rarity $r_c$, Difficulty $d_c$    *(Eq. 15)*
3: Base Score $S_{t,c} \leftarrow A_{t,c} +$ Advantage $+$ Specialist    *(Eq. 16)*
4: Boost $B_c \leftarrow 1 + \beta r_c d_c \mathbf{1}_{tail}$    *(Eq. 17)*
5: Normalize $w'_{t,c} \leftarrow \mathrm{Softmax}(S_{t,c} \cdot B_c / \tau_c)$
6: Update $w^{(new)}_{t,c} \leftarrow m_c w^{(old)}_{t,c} + (1 - m_c) w'_{t,c}$    *(Eq. 18)*
7: **— Phase B: Inference Arbitration (Testing) —**
**Require:** Input $x_i$, Task Logits $\{\mathbf{z}^{(t)}\}$, Weights $w_{t,c}$
8: Get Predictions $\hat{y}_t = \arg\max \mathbf{z}^{(t)}$
9: **if** All $\hat{y}_t$ are identical **then**
      Soft fusion of probabilities
10: **else**
11:     **for** each task $t$ **do**
12:         Conf $\mathcal{C}_{t,i} \leftarrow \mathrm{Softmax}(\mathbf{z}^{(t)})[\hat{y}_t]$
13:         Margin $M_{t,i} \leftarrow (\mathrm{Top1} - \mathrm{Top2})/\mathrm{Top1}$
14:         $S_{t,i} \leftarrow \mathcal{C}_{t,i}(1 + \lambda w_{t,\hat{y}_t}) + \mu M_{t,i}$    *(Eq. 19)*
15:     **end for**
16:     Best Task $t^* \leftarrow \arg\max_t S_{t,i} \; \mathbf{p}^{(t^*)}$ (or weighted fusion)
17: **end if**

---

# F. Limitations

While our proposed AES framework demonstrates superior performance in long-tailed recognition, we acknowledge several limitations that warrant future investigation:

- **Computational Overhead:** The *Entropy-aware PCGrad (E-PCG)* requires computing pairwise cosine similarities and entropy measures for task gradients. Although effective for resolving conflicts, this process introduces additional computational cost and memory usage during the training phase compared to standard scalarization methods, particularly when the number of task objectives or model parameters is extremely large.

- **Hyperparameter Complexity:** To achieve fine-grained control over the optimization dynamics, our method introduces specific hyperparameters (e.g., the temperature $\tau$ in ARS and the momentum coefficients in SCAF). While our sensitivity analysis demonstrates robustness across a reasonable range, deploying the framework on datasets with vastly different distributions may still require heuristic tuning to find the optimal configuration.

- **Dependency on Validation Statistics:** The *Sample-level Conflict Arbitrated Fusion (SCAF)* relies on class-wise priors (accuracy and frequency) estimated from a held-out validation set. In scenarios where the validation set is extremely small or not representative of the test distribution (e.g., severe distribution shift), the estimated priors may be biased, potentially affecting the precision of the inference-time routing.

- **Centralized Paradigm Constraint:** AES is currently limited to centralized training. In decentralized or federated applications, statistical heterogeneity (Non-IID data) across nodes can further aggravate optimizer bias. Extending our state-aware corrections to cooperate with distributed protocols like GDAP (Xu et al., 2026) remains an open future direction.

