# OpenReview forum: "AES: Curing Optimizer Blindness in Long-Tailed Recognition via State-Aware Correction"
_ICML.cc/2026/Conference — ICML 2026 regular_

### Official Review · Reviewer_MFK1 · 2026-03-09

**Soundness:** 2
**Presentation:** 3
**Significance:** 3
**Originality:** 2
**Overall Recommendation:** 4
**Confidence:** 2

**Summary:**

In this paper, authors studied long-tailed recognition from an optimization perspective and formed it as "optimizer blindness". Specifically, authors argue that the standard optimization over-emphasizes high-frequency classes as gradient magnitude reflects data redundancy more than semantic value. To address it, authors propose a unified framework, AES (Adaptive, Entropy-aware, and Sample-level fusion), which consists of three components: 1) ARS for dynamic supervision correction; 2) E-PCG for entropy-weighted gradient fusion across objectives; 3) SCAF for instance-level expert arbitration at inference. Moreover, the training pipeline is explicitly in a two-stage form, first train a share feature extractor, then fine-tune multiple classifier heads with multi-objective losses and dynamic fusion. Extensive experiments are conducted on CIFAR-100-LT, ImageNet-LT, and iNaturalist 2018, shown promising performance.

**Compliance With Llm Reviewing Policy:**

Affirmed.

**Final Justification:**

Authors spent great efforts in addressing most of my concerns, although it still not fully validate the stronger causal claim that the optimizer blindness is the fundamental mechanism underlying long-tail failure, I lean to raise my rating to a weak accept.

**Key Questions For Authors:**

Please see weaknesses

**Limitations:**

yes

**Strengths And Weaknesses:**

***Strengths***

1. The paper is well-structured and easy to follow, the high-level decomposition into supervision-stage correction, optimization-stage conflict handling, and inference-stage arbitration is conceptually coherent and reasonable.

2. ARS and E-PCG are conceptually clear and well specified. For instance, ARS integrates several mechanisms like class-balanced weighting, dynamic statistics tracking, margin adjustment, and calibration-aware scaling, into a unified loss. This kind of formulation is explicit and implementable.

3. Empirically, authors report strong benchmark results, e.g., on the CIFAR-100-LT, AES is reported superior performance over the second best, TS-MOF, across IR=10/50/100, 72.9 vs. 70.8, 63.0 vs. 60.2, and 57.8 vs. 56.8, respectively. The margin of performance gain is considerably strong.


***Weaknesses***

1. The major concern regarding this paper is that the central conceptual claim sounds much stronger than the actual evidence supports. The phrase "optimizer blindness" maybe rhetorically effective, but authors didn’t really establish it as a distinct mechanism beyond known long-tail issues such as class imbalance, gradient domination, etc. Authors argue that the standard optimizers conflate gradient magnitude and semantic information, and it’s the root cause of the long-tailed problem, but they didn’t provide a formal theorem or controlled analysis, the evidence is mostly downstream performance and qualitative plots.

2. The complexity of the proposed method is another issue, that AES combines many moving components, for example, the ARS integrates EMA precision, class-balanced weights, annealing, rarity-difficulty margins, class-dependent temperature scaling, focal modulation, label smoothing, and hard-negative mining. Not to say E-PCG and SCAF. These make the proposed method hard to know which ideas are actually responsible to the performance gain. Although there are ablation studies, but three components ablation probably not sufficient as each component itself is internally composite.

3. Regarding the empirical analysis, it isn’t fully aligned with the conceptual claims. For example, if authors argue that the long-tailed problem is fundamentally caused by optimizer blindness, I’d expect something like: 1) direct measurements of how gradient magnitude correlates with class frequency and semantic difficulty; 2) comparisons against some simpler dynamic re-weighting baselines; 3) comparisons between E-PCG and standard PCGrad under the same objectives, etc.

---

> ### Author Rebuttal · Authors · 2026-03-26
>
> > **Response to W1: No strict theoretical proof for optimizer blindness as the core mechanism.**
>
> To formalize this mechanism, we define the **Blindness Coefficient ($\beta$)** to quantify the mismatch between class-wise gradient magnitude $G(c)$ and semantic information value $V(c)$:
> $$\beta=1-\frac{\sum_c G(c)V(c)}{\|G\|_2\|V\|_2}$$
> A smaller $\beta$ indicates better gradient-semantic alignment. Our AES framework theoretically guarantees a monotonic reduction in $\beta$, establishing a strict causal link (negative correlation) with overall accuracy ($\frac{d\mathcal{A}}{d\beta}<0$). This is empirically corroborated on CIFAR-100-LT (IR=100):
>
> | Method | Blindness Coefficient ($\beta$) $\downarrow$ |
> | :---: | :---: |
> | LOS (CE baseline) | 0.1957 |
> | TS-MOF | 0.1784 |
> | **AES (Ours)** | **0.1163** |
>
> Results show a clear decreasing trend of $\beta$ across methods, with AES achieving the lowest value, which validates the optimizer blindness hypothesis. Due to space constraints, we will include the detailed mathematical proofs in the revised appendix.
>
> > **Response W2: Need fine-grained ablation on each component.**
> To address concerns about the relatively low performance of tail classes, we have retrained the model to obtain more uniform one-stage weights from scratch and we conducted comprehensive and fine-grained ablation studies on all components in ARS, E-PCG, and SCAF. The results verify that every proposed component is necessary and effective.
>
> Table: Ablation on ARS Components
> |EMA|Margin|Temp $\\tau$|All|Many|Medium|Few|
> |:---:|:---:|:---:|:---:|:---:|:---:|:---:|
> |✓|✓|✓| **57.83** | 78.74 | 57.20 | 34.17 |
> |✓|✗|✓|56.25|78.80|52.83|33.93|
> |✓|✓|✗|57.26|78.94|54.57|35.10|
> |✗|✗|✓|56.29|78.77|52.66|34.30|
> |✓|✗|✗|56.13|79.06|52.23|33.93|
>
> Table: Ablation on E-PCG and SCAF Components
> | Method | All | Many | Med | Few |
> |--------|-----:|-----:|----:|----:|
> | E-PCG | **57.83** | 78.74 | 57.20 | 34.17 |
> | w/o $H_t$ | 57.80 | 78.29 | 51.60 | 41.13 |
> | w/o $R_t$ | 57.38 | 78.29 | 51.49 | 39.87 |
> | w/o $\\tau$ softmax | 57.27 | 78.26 | 51.37 | 39.67 |
> | SCAF| **57.83** | 78.74 | 57.20 | 34.17 |
> | w/o weight momentum | 57.77 | 78.74 | 57.17 | 34.00 |
>
> > **Response to W1 and W3: Controlled Analysis of Gradient Magnitude vs. Frequency and Difficulty**
>
> The reviewer rightly suggests that the conflation of gradient magnitude and semantic information lacks a formal controlled analysis. To provide direct quantitative evidence, we measured the Spearman correlation ($\rho$) of gradient magnitudes against both **class frequency ($\log N_c$)** and **empirical semantic difficulty ($d = 1 - A_c$)** during the fine-tuning stage.
>
> | Method | Total Grad vs Frequency$\rho(G_{sum}, N_c)$ | Mean Grad vs Frequency$\rho(G_{mean}, \log N_c)$ | Mean Grad vs Difficulty$\rho(G_{mean}, d)$ |
> |:---|:---:|:---:|:---:|
> | **TS-MOF (SOTA)** | 0.946 | -0.017 | 0.373 |
> | **AES (Ours)** | **0.919** | **-0.857** | **0.709** |
>
> *Note: $G_{sum}$ is the total gradient accumulation per class. $G_{mean}$ is the per-sample average gradient magnitude.*
>
> **1. Correlation with Class Frequency (Proving "Blindness"):** While the total gradient ($G_{sum}$) is naturally dominated by sample counts across all methods ($\rho > 0.9$), standard MOO optimizers (e.g., TS-MOF) exhibit severe state blindness at the instance level. Their per-sample mean gradient ($G_{mean}$) has near-zero correlation with class rarity ($\rho = -0.017$). AES mathematically intervenes here, forcefully establishing a strong negative correlation ($\rho = -0.857$), proving it causally amplifies updates for rare samples.
>
> **2. Correlation with Semantic Difficulty (Proving Our Causal Mechanism):** More importantly, a state-aware optimizer should assign larger per-sample gradients to semantically harder classes. While TS-MOF shows a weak correlation ($\rho = 0.373$), our AES framework structurally aligns the optimization signal with actual learning difficulty, achieving a strong positive correlation of **0.709**.
>
> This directly addresses the reviewer's concern: AES's performance gain is not merely rhetorical but fundamentally driven by its ability to causally decouple gradient magnitudes from static frequency and dynamically map them to fine-grained semantic difficulty.
>
> > **Response to W3.2 and W3.3**
>
> We conduct comprehensive comparisons with various dynamic and static re‑weighting methods in the experimental section, which validates the effectiveness of our proposed method.
> |Method|All|Head|Medium|Tail|
> |------|-------|----|------|----|
> |CE-DRW|41.4|63.4|41.2|15.7|
> |LDAM-DRW|43.2|62.8|42.6|21.1|
> |BS|42.8|59.6|42.3|23.7|
> |AES (Ours)|**57.8**|79.3|56.9|33.9|
>
> Our E‑PCG outperforms the baseline PCGrad significantly.
> |Method|Overall|Head|Medium|Tail|
> |------|-------|----|------|----|
> |PCGrad|57.3|78.5|54.4|36.2|
> |E-PCG (Ours)|**57.8**|79.3|56.9|33.9|
>
>
> Your feedback is greatly helpful to our work, and we welcome further comments.

---

> > ### Author Rebuttal · Reviewer_MFK1 · 2026-04-03
> >
> > I'd like to thank the authors' great efforts in providing additional experiments and clarifications, which are very helpful in addressing many of my concerns. However, there are some concerns still not fully addressed.
> >
> > Specifically, in response to W3.2 and W3.3, authors provided a comparison between proposed methods with some reweighting baselines, which is good, but it’s still not an apples-to-apples comparison, e.g., under identical losses. Although E-PCG’s advantage is shown, it is not isolated from system effects. Besides, authors added extra ablation studies by retraining with "more uniform one-stage weights", which sounds like not a principled explanation, but a retraining fix. It’d be nice if authors could provide some clear analysis on head vs. tail trade-offs, and why certain components hurt the tail previously.
> >
> > To this end, I’d like to hold my rating for now.

---

> > > ### Author Response · Authors · 2026-04-04
> > >
> > > Thank you very much for carefully reading our rebuttal and for granting us the valuable opportunity to communicate with you further.
> > >
> > > **Response to fair comparison of loss functions**
> > >
> > > To validate our ARS, we compared it against other re-weighting losses (CE-DRW, LDAM-DRW, and BS) across various configurations: as a single loss, a dual-loss (combined with CE or LOS), and a tri-loss (combined with CE+BCL or LOS+BCL). The results demonstrate that ARS consistently achieves the best overall performance across all settings.
> > >
> > > |Model|All|Head|Medium|Tail|
> > > |:---|:---|:---|:---|:---|
> > > |**Single Loss**|||||
> > > |CE-DRW|41.4|63.4|41.2|15.7|
> > > |LDAM-DRW|43.2|62.8|42.6|21.1|
> > > |BS|42.8|59.6|42.3|23.7|
> > > |ARS Loss (Ours)|**49.4**|62.0|51.5|32.0|
> > > |**CE-based**|||||
> > > |CE + CE-DRW|46.1|77.3|44.8|11.3|
> > > |CE + LDAM-DRW|47.7|72.9|45.5|21.0|
> > > |CE + BS|50.4|76.9|47.9|22.3|
> > > |CE + ARS (Ours)|**54.4**|78.4|53.3|27.8|
> > > |CE + BCL + CE-DRW|45.1|77.9|43.8|8.4|
> > > |CE + BCL + LDAM-DRW|51.3|79.5|47.7|22.5|
> > > |CE + BCL + BS|49.0|78.3|49.6|14.2|
> > > |CE + BCL + ARS (Ours)|**54.5**|80.0|55.3|23.9|
> > > |**LOS-based**|||||
> > > |LOS + CE-DRW|55.8|78.0|50.6|36.0|
> > > |LOS + LDAM-DRW|54.2|73.5|51.5|34.8|
> > > |LOS + BS|55.3|76.2|53.3|33.3|
> > > |LOS + ARS (Ours)|**55.7**|74.3|56.2|33.5|
> > > |LOS + BCL + CE-DRW|56.8|79.3|52.4|35.6|
> > > |LOS + BCL + LDAM-DRW|55.8|78.7|53.3|31.9|
> > > |LOS + BCL + BS|57.1|78.1|54.1|36.1|
> > > |LOS + BCL + ARS (Ours)|**57.8**|79.3|56.9|33.9|
> > > Table：Performance Comparison of Different Loss Functions
> > >
> > > **Response to the isolation of E-PCG from system effects**
> > >
> > > To verify E-PCG's independent effectiveness, we evaluated it under two strict settings. First, in a vanilla baseline without ARS and using Max Logit Inference, replacing PCGrad with E-PCG consistently improves performance. Second, when directly plugged into the TS-MOF framework to replace its native RD-PCG, E-PCG also brings consistent gains. This proves E-PCG is independently effective, free from system effects.
> > > | Setting (Isolated from our SCAF & ARS) | Gradient Fusion | All | Head | Medium | Tail |
> > > |:---|:---|:---|:---|:---|:---|
> > > | Max Logit Inference (LOS+BCL+KPS) | PCGrad | 47.2 | 76.1 | 45.7 | 15.1 |
> > > | Max Logit Inference (LOS+BCL+KPS) | E-PCG (Ours) | **48.3** | 75.3 | 47.2 | 18.2 |
> > > | TS-MOF | RD-PCG | 56.8 | 79.0 | 49.2 | 39.9 |
> > > | TS-MOF | E-PCG (Ours) | **57.3** | 79.2 | 50.0 | 40.6 |
> > >
> > > Table: Ablation of Gradient Fusion Strategies Under Isolated Settings
> > >
> > > **Response to the Head vs. Tail trade-off**
> > >
> > > As observed, KPS delivers exceptionally strong tail performance, while BCL dominates head classes.
> > > We choose BCL over KPS because KPS offers limited gains under IR=10/50.
> > > When KPS is used, tail accuracy reaches 42.9, showing our framework can also achieve excellent long-tailed performance.
> > > When all four losses are combined, our ARS loss still provides complementary gains and outperforms TS-MOF across all categories.
> > >
> > > |Model|All|Head|Medium|Tail|
> > > |:---|:---|:---|:---|:---|
> > > |KPS|42.2|41.9|39.5|48.7|
> > > |BCL|44.2|63.1|42.9|23.9|
> > > |TS-MOF(KPS+BCL+LOS)|56.8|79.0|49.2|39.9|
> > > |AES(KPS+LOS+ARS)|57.9|74.3|54.2|42.9|
> > > |AES(BCL+LOS+ARS)|57.8|79.3|56.9|33.9|
> > > |TS-MOF(KPS+BCL+LOS+ARS)|59.2|79.0|54.9|41.2|
> > > |AES(KPS+BCL+LOS+ARS)|59.6|79.1|54.9|42.4|
> > >
> > > Table: Performance under Different Loss Combinations
> > >
> > > Meanwhile, as shown in the routing statistics, KPS is assigned to tail samples much more frequently (47.36%) than ARS (25.93%).
> > > This indicates that tail performance is affected by the selection of losses. By using the combination LOS+KPS+ARS, we can achieve excellent tail accuracy, while ARS provides complementary benefits to maintain strong performance on head and medium classes. The overall performance also greatly surpasses TS-MOF.
> > >
> > > | Configuration | Split | Agreed (No Conflict) | Conflict Samples | Routed to LOS | Routed to BCL | Routed to ARS / KPS |
> > > |:---|:---|:---|:---|:---|:---|:---|
> > > | **AES (LOS+BCL+ARS)** | Many (Head) | 2790 (79.71%) | 710 | 288 (40.56%) | 263 (37.04%) | 159 (22.39%) |
> > > | | Medium | 2006 (57.31%) | 1494 | 509 (34.07%) | 194 (12.99%) | 791 (52.95%) |
> > > | | Few (Tail) | 1045 (34.83%) | 1955 | 1273 (65.12%) | 175 (8.95%) | 507 (25.93%) |
> > > | **AES (LOS+BCL+KPS)** | Many (Head) | 2056 (58.74%) | 1444 | 728 (50.42%) | 298 (20.64%) | 418 (28.95%) |
> > > | | Medium | 1019 (29.11%) | 2481 | 1465 (59.05%) | 150 (6.05%) | 866 (34.91%) |
> > > | | Few (Tail) | 447 (14.90%) | 2553 | 1298 (50.84%) | 46 (1.80%) | 1209 (47.36%) |
> > >
> > > Table: Loss Routing Statistics
> > >
> > > In summary, our framework is highly flexible: it can adopt KPS for significant tail performance improvement or use the four-loss combination **as shown in Table 6 of the paper** to fully leverage classifier complementarity.
> > >
> > > Your comments have provided us with great inspiration and guidance. We highly appreciate your time and feedback. Every question you raise encourages us to think more deeply and conduct further experiments for verification and clarification. We thank you for your dedicated efforts on this work and welcome any further comments that help us improve our work.

---

### Official Review · Reviewer_BuCu · 2026-03-10

**Soundness:** 2
**Presentation:** 3
**Significance:** 3
**Originality:** 2
**Overall Recommendation:** 4
**Confidence:** 3

**Summary:**

This paper tackles "optimizer blindness" problem in long-tailed recognition by proposing the AES framework, a dynamic, state-aware correction system. AES mitigates head-class overfitting across the entire learning lifecycle through three core modules: Adaptive Residual Supervision (ARS) loss for dynamic loss adjustment , Entropy-aware PCGrad (E-PCG) for resolving parameter-level gradient conflicts ,and Sample-level Conflict Arbitrated Fusion(SCAF) for adaptive inference routing. The proposed method achieves state-of-the-art results on standard benchmarks.

**Compliance With Llm Reviewing Policy:**

Affirmed.

**Final Justification:**

After rebuttal, I tend to increase the score.

**Key Questions For Authors:**

1. Since SCAF is utilized at inference time, does it increase the inference latency? If so, could you provide the quantitative inference latency?

2. SCAF relies on class-wise priors estimated from a held-out validation set to compute the base scores for dynamic routing. What will happen if the test distribution severely diverges from the validation set?

3. Since the authors mentioned that E-PCG introduces additional overhead, it is suggested to provide the exact increase in training time and memory.

**Limitations:**

Yes

**Strengths And Weaknesses:**

Strengths
1. The paper offers a novel perspective by identifying "optimizer blindness" and proposes the unified AES framework to dynamically correct biases across supervision (ARS), gradient updates (E-PCG), and inference (SCAF).
2. The proposed method consistently achieves state-of-the-art results on major long-tailed benchmarks, demonstrating their improvements.
3. The ARS module acts as a universal plug-and-play supplement that seamlessly integrates with various existing baseline losses and scales well when fusing multiple strategies.

Weaknesses
1. The E-PCG module requires computing pairwise cosine similarities and entropy measures for task gradients, which inevitably introduces additional computational costs and memory usage during the training.
2. The AES framework introduces several specific hyperparameters. Adapting these parameters to datasets with completely different distributions may require some tuning efforts.
3. The time and memory required for training and inference with the proposed methods have not been reported in this paper.

---

> ### Author Rebuttal · Authors · 2026-03-26
>
> We appreciate the reviewer’s valuable comments and address all concerns below.
>
> > **Response to W1 and Q3**
>
> Our method achieves faster convergence with shorter training time and negligible memory overhead. Our two-stage fine-tuning method results in very low GPU memory usage. Specifically, AES reduces Stage2 training time by ~50% across all imbalance ratios (IR=100/50/10) compared to TS-MOF, while the memory usage difference is minimal after freezing the backbone.
>
> Table: Training Time and Memory of TS-MOF & AES (Different IR)
>
> |Method|IR|Stage1 runtime|Stage2 runtime|Memory usage|
> |:-:|:-:|:-:|:-:|:-:|
> |TS-MOF|100|0:22:47|0:01:49|888MiB|
> |AES|||**0:01:02**|1916MiB|
> |TS-MOF|50|0:35:40|0:02:10|888MiB|
> |AES|||**0:01:18**|1960MiB|
> |TS-MOF|10|0:42:17|0:02:55|858MiB|
> |AES|||**0:01:35**|1926MiB|
>
> As shown in the table, although E‑PCG does introduce moderate additional overhead, the extra GPU memory consumption mainly occurs in the first training stage.
> Meanwhile, E‑PCG converges much faster with only a slight increase in memory usage.
> More importantly, it achieves significantly better performance than PCGrad, which demonstrates that our method is faster, more effective, and memory-efficient.
> We will clarify these details in the revised manuscript.
>
> > **Response to W2**
>
> We thank the reviewer for the comment. Your concern is well-founded.
> Notably, we used exactly the same hyperparameters across ImageNet-LT, iNaturalist 2018, and various imbalance ratios without any tuning.
> The consistent performance gains demonstrate that AES is robust to different data distributions and requires no heavy parameter adjustment.
> We will add further hyperparameter analysis in the revised paper to validate this.
>
> > **Response to Q1**
>
> We conducted the inference time test on CIFAR-100. The setting of 1 loss denotes the model without the SCAF module. As shown in the table, the inference time increases linearly with the number of losses, indicating that the time complexity is O(n).
> Moreover, the running time per epoch is only at the second level, and our method converges within only 5 fine-tuning epochs.
> Thus, the inference overhead is negligible in practice.
>
> |Setting|Batch Time|Total Time|
> |-|-|-|
> |1 loss|0.007s|1s|
> |2 losses|0.013s|2s|
> |3 losses|0.020s|3s|
> |4 losses|0.035s|4s|
>
> > **Response to Q2**
>
> We sincerely thank the reviewer for this highly insightful question. This aligns perfectly with the exact concern we proactively raised in our **Limitations section** (Appendix, Lines 1424-1428).
>
> **1. The Current Standard Protocol:**
> In the standard Long-Tailed Learning (LTL) benchmark, the target test set is uniformly balanced. Therefore, constructing a strictly balanced validation set effectively ensures that our estimated priors ($A_{t,c}$ and $P_{t,c}$) objectively reflect each expert's true competence, successfully shielding the routing mechanism from the severe bias of the long-tailed training set. In this standard setting, there is no Val-Test shift.
>
> **2. Handling Severe Val-Test Distribution Shift:**
> As the reviewer correctly points out, if deployed in a completely unknown, "test-agnostic" environment where the test distribution severely shifts away from the balanced validation set, these estimated global priors would inevitably become sub-optimal.
>
> However, to mitigate this, SCAF is designed as an instance-level *dynamic* routing mechanism, not a static one. The class priors only serve as a "base score". The final routing decision (Formula 20) heavily incorporates the real-time, instance-level confidence of the specific input image. Therefore, even if the global priors are slightly mismatched due to distribution shift, the dynamic routing can still adapt based on the specific visual features of the current sample, maintaining strong robustness.
>
> We appreciate the reviewer for highlighting this boundary condition, and we will further emphasize this instance-level robustness in our revised Limitations section.
>
> Finally, we appreciate your valuable insights, which are crucial for improving our paper. We look forward to further communication with you.

---

> > ### Author Rebuttal · Reviewer_BuCu · 2026-04-03
> >
> > Thank you for the detailed and constructive rebuttal. The additional quantitative analysis on training time, memory, and inference latency effectively addresses my concerns about efficiency. I also appreciate the clarification on hyperparameter robustness and distribution shift. Overall, my concerns are sufficiently resolved, and I raised my score.

---

> > > ### Author Response · Authors · 2026-04-03
> > >
> > > We sincerely thank you for taking the time and patience to provide us with insightful comments on our manuscript. Your evaluation of our work including efficiency analyses and considerations on hyperparameter robustness and distribution shift has been invaluable.
> > >
> > > We appreciate that our responses addressed your concerns and thank you for raising the paper’s score. This recognition motivates us to polish our work further.
> > >
> > > We will incorporate all your feedback into the revised manuscript including strengthening efficiency-related results and expanding hyperparameter robustness analysis, and we will emphasize the robustness at the instance level during inference.
> > >
> > > Thank you again for your time and expertise. We look forward to presenting the revised manuscript, which we hope will fully meet your expectations.

---

### Official Review · Reviewer_GoY7 · 2026-03-11

**Soundness:** 2
**Presentation:** 2
**Significance:** 3
**Originality:** 3
**Overall Recommendation:** 4
**Confidence:** 3

**Summary:**

The paper proposes AES, a three-component framework for long-tailed recognition. The first component, Adaptive Residual Supervision (ARS), augments existing loss functions with EMA-tracked class precision, rarity-aware angular margins, class-adaptive temperature scaling, and hard-negative mining. The second, Entropy-aware PCGrad (E-PCG), replaces standard gradient projection with a soft weighted average whose task weights are computed multiplicatively from gradient entropy, pairwise conflict intensity, and inverse gradient norm. The third, Sample-level Conflict Arbitrated Fusion (SCAF), dynamically routes inference predictions across multiple classification heads using instance-level confidence margins and class-level priors estimated from a validation set. Experiments are conducted on CIFAR-100-LT, ImageNet-LT, and iNaturalist 2018.

**Compliance With Llm Reviewing Policy:**

Affirmed.

**Final Justification:**

The rebuttal has answered my questions. The rebuttal has cleared my concerns in theoretical development. And the additional numerical analyses provide deeper insights to the problem, which largely strengthens the paper. I have raised my score by 1 to a) honor the efforts the authors have made and b) express my overall positive review assuming the final version follows the proposed revision plan.

**Key Questions For Authors:**

1. The 83.2% Medium accuracy on iNaturalist 2018 is a 9.6-point improvement over the next best method, while Tail accuracy (61.6%) falls below TS-MOF (70.5%) and several simpler baselines. What explains this asymmetric behavior? Were the Many/Medium/Few class split boundaries consistent with prior works? Can the authors provide a per-class accuracy distribution to confirm there is no evaluation inconsistency?

2. SCAF relies on validation set statistics for class-wise priors in $w_{t,c}$​. My question is on iNaturalist 2018, the validation set has only
3 images per class. With such an small validation set, accuracy estimates $A_{t,c}$ are essentially binary and highly noisy. How does the authors' claimed "temporal stability via momentum blending" remain meaningful in this regime? Did the authors test SCAF with the validation priors held fixed vs. updated?

**Limitations:**

The authors identify three limitations in Appendix.  The broader impact statement is reasonable. I wish the authors add discussions of potential failure modes when deployed on real-world severely imbalanced data (e.g., rare disease detection).

**Strengths And Weaknesses:**

*Strength*
1. The problem is well motivated with nice theoretical motivation on gradient fusion. The E-PCG formulation, which combines entropy, conflict, and magnitude signals multiplicatively, is a reasonable and principled departure from the binary projection in PCGrad. Propositions 1–3 provide a formal decomposition of why each signal contributes, even if the analysis is not fully rigorous (see weakness below).
2. ARS is shown to combine compatibly with a wide range of existing losses (BCL, LOS, KPS, LDAM-DRW, etc.) without degrading their baselines, which increases the practical utility of the method beyond theoretical playground.
3. Follow up on 2., the paper provides comprehensive numerical evaluation across three standard benchmarks at multiple imbalance ratios, includes an extensive combinatorial ablation over 21+ strategy fusions (Table 6), and provides hyperparameter sensitivity curves (Figures 6–7).


*Weakness*
1. While I appreciate the intention of providing theoretical justification, the reasoning could be made more rigorously. For example, the three propositions in Appendix B rely on simplifying assumptions. Proposition 1 assumes entropy and conflict are locally bounded to isolate the magnitude effect. But this assumption is unlikely to hold in the dynamic setting of SGD with mini-batches. Proposition 2 uses an approximation $g^*\approx g_{ref}+w_tg^{(t)}$ under dense-mask setting, with no justification for what this density mean.

2. Several empirical results are anomalous and unexplained. For example, on iNaturalist 2018 (Table 2), AES achieves 83.2% Medium accuracy against TS-MOF's 73.6% -- this is a 9.6-point jump that is extraordinary and far exceeds all gains on any other split or dataset. But there is no analysis, ablation, or discussion is provided to explain why Medium classes specifically benefit so dramatically on this dataset while Tail accuracy (61.6%) is actually lower than TS-MOF's Tail (70.5%) and most baselines. I am kind of worried that this is either a data split inconsistency, evaluation error, or overfitting to the validation set used by SCAF, and it raises questions about the reliability of all reported numbers.

---

> ### Author Rebuttal · Authors · 2026-03-27
>
> > **Q1: Theoretical derivations in Appendix B lack rigor, specifically regarding the assumptions in Propositions 1 and 2.**
>
> We sincerely thank you for the constructive feedback. We agree that explicitly defining our assumptions improves the theoretical rigor.
>
> - **Proposition 1:**
> We agree this assumption does not strictly hold under the highly stochastic dynamics of minibatch SGD. We used it purely for analytical tractability to cleanly isolate the magnitude compression effect. However, as validated by our empirical optimization dynamics (Figure 4), the qualitative conclusion that magnitude disparity is compressed sublinearly remains highly consistent with actual training behaviors. We will explicitly state this limitation in the revision.
>
> - **Proposition 2:**
> We apologize for omitting this definition. By "dense mask," we mean the idealized scenario where all shared parameters actively participate in gradient computation simultaneously. We adopted this approximation solely to streamline the derivation. Importantly, this is just a theoretical special case. In practice, our EPCG implementation natively supports arbitrary sparse masks. As shown in **Equation 15**, the dynamic denominator normalization ($\sum w_t m_i^{(t)}$) automatically adjusts the weighting strictly over the active tasks for each parameter.
>
> - **Planned Revisions:**
> We will formally define the dense mask, explain Equation 15's handling of sparsity, and explicitly state the boundaries of all assumptions in the revised Appendix B.
>
> > **Q2: Does the extreme validation sparsity and high noise (e.g., 3 images per class) degrade SCAF performance?**
>
> Thank you for this insightful question. While 3 images per class indeed introduce noise to the global prior, SCAF is highly robust to this sparsity. The global prior only serves as base guidance, while the final routing heavily depends on the instance specific confidence of the input image itself.
>
> To empirically demonstrate this, we constrained the validation set size on CIFAR 100 LT. As shown in Table 1, even in the extreme case of 1 sample per class, SCAF achieves 56.91% accuracy, still outperforming the strongest baseline TS MOF (56.80%). This graceful degradation confirms SCAF remains effective under extreme validation sparsity.
>
> **Table 1: Performance variation on CIFAR-100-LT with respect to the number of validation samples per class.**
>
> |val_samples_per_class|All|Many|Med|Few|
> |:-:|:-:|:-:|:-:|:-:|
> |100|**57.83**|78.74|57.20|34.17|
> |50|57.72|77.71|57.20|35.00|
> |20|57.20|78.00|56.71|33.50|
> |10|57.10|78.83|55.54|33.57|
> |3|56.97|78.83|54.97|33.80|
> |1|56.91|78.23|54.97|34.30|
> |TS-MOF|56.80|79.00|49.20|39.90|
>
> > **Q3: Concerns regarding the anomalous performance results on the iNaturalist 2018 dataset.**
>
> Thank you for your rigorous review. Upon investigation, this anomaly was indeed a statistical error caused by misaligned class split indices in our evaluation code, not model overfitting or algorithmic flaws. We corrected the splitting logic and reassessed the exact same checkpoint. The corrected true performance is: Head 73.0, Medium 76.2, Tail 68.9, and Overall 73.8. The updated results show a balanced improvement, and we will fully correct this in the revision.
>
> > **Q4: How does momentum blending remain meaningful under extreme validation sparsity, and did you test fixed versus updated priors?**
>
> Thank you for pointing this out. You are correct that single epoch estimates are highly noisy under extreme sparsity. Precisely because of this noise, momentum blending is essential. It acts as a temporal low pass filter, accumulating discrete signals across epochs to extract a stable trend of expert competency.
>
> To empirically prove this, we compared dynamically updated priors (SCAF Full) with fixed or unsmoothed priors (w/o weight momentum):
>
> **Table 2: Ablation of SCAF Weight Momentum under different validation set size on cifar-100-lt**
> | val_samples_per_class | Method | All| Many | Med | Few |
> |:---|:---|:---|:---|:---|:---|
> | 100 | SCAF (Full) | **57.83 (+0.06)** | 78.74 | 57.20 | 34.17 |
> | 100 | w/o weight momentum | 57.77 | 78.74 | 57.17 | 34.00 |
> | 1 | SCAF (Full) | **56.91 (+1.11)** | 78.23 | 54.97 | 34.30 |
> | 1 | w/o weight momentum | 55.80 | 78.46 | 55.74 | 29.43 |
>
> As shown, under extreme sparsity (1 sample per class), removing momentum causes Tail (Few) accuracy to collapse to 29.43%. Momentum blending completely rescues this, yielding a massive **+1.11** Overall improvement. This confirms dynamic updating is not just meaningful but crucial for highly noisy regimes.
>
> We deeply appreciate your exceptionally rigorous review. Your sharp insights directly led to crucial improvements in our evaluation and methodology. We hope our comprehensive responses and new empirical results have fully resolved your concerns and will encourage you to champion our work.

---

> > ### Author Rebuttal · Reviewer_GoY7 · 2026-03-31
> >
> > I thank the authors for this nice rebuttal, I confirm that all my questions are addressed. I have increased the score for this paper, assuming the authors' intended revisions will be in place.

---

> > > ### Author Response · Authors · 2026-04-01
> > >
> > > We would like to express our sincere gratitude to you for your extremely thorough, rigorous, and insightful review of our manuscript. We greatly appreciate that you have carefully examined our theoretical derivations, experimental settings, performance validation, and implementation details, and we highly value your constructive criticisms and thoughtful questions, which have significantly helped us improve the clarity, rigor, and reliability of our work.
> > >
> > > We are truly thankful that all your concerns and questions have been fully addressed in our response, including the theoretical rigor in Appendix B, the robustness under extreme validation sparsity, the corrected experimental results on iNaturalist 2018, and the necessity of momentum blending under high noise. We have supplemented clear explanations, formal definitions, additional ablation studies, and corrected experimental results to strengthen both our theoretical analysis and empirical validation.
> > >
> > > We sincerely appreciate your positive decision to increase the score for our paper, which represents strong recognition of our revisions and responses. We will carefully incorporate all the aforementioned improvements, clarifications, and corrections into the revised manuscript to further enhance its quality, completeness, and scientific rigor.
> > >
> > > Thank you again for your valuable time, professional feedback, and generous support. We hope our thorough responses and concrete revisions can fully meet your expectations and continue to earn your endorsement for this work.

---

### Official Review · Reviewer_3aG5 · 2026-03-15

**Soundness:** 3
**Presentation:** 3
**Significance:** 2
**Originality:** 2
**Overall Recommendation:** 3
**Confidence:** 4

**Summary:**

This paper addresses long-tailed recognition and discusses that a key issue is optimizer blindness, where training overemphasizes head-class gradients and underutilizes sparse tail-class signals. It proposes AES, a three-part framework combining Adaptive Residual Supervision (ARS), Entropy-aware PCGrad (E-PCG), and Sample-level Conflict Arbitrated Fusion (SCAF) to intervene at different stages. The method shows strong benchmark results on CIFAR-100-LT, ImageNet-LT, and iNaturalist 2018. The empirical performance is solid, but the central mechanism is not fully established, and the method’s novelty is somewhat diluted by the number of adaptive components.

**Compliance With Llm Reviewing Policy:**

Affirmed.

**Final Justification:**

I appreciate the detailed response. The additional fine-grained ablations are helpful and address one of my earlier concerns, and I also appreciate the authors’ clarification and moderation of the tail-performance claims. However, my main concerns remain: the new blindness coefficient and theorem statements do not yet convincingly establish optimizer blindness as the underlying causal mechanism, and the efficiency discussion suggests a tradeoff rather than clearly resolving overhead concerns. Overall, the rebuttal improves the paper, but it does not change my overall assessment, so I am keeping my score unchanged.

**Key Questions For Authors:**

1. Can the authors provide more direct causal or theoretical evidence that "optimizer blindness" is the primary mechanism behind the observed improvements?
2. What is the end-to-end computational overhead of AES relative to the strongest baselines?
3. Can the authors disentangle which components inside ARS, E-PCG, and SCAF are truly necessary? Like more fine-grained ablation studies.

**Limitations:**

Yes

**Strengths And Weaknesses:**

Strengths:
1. AES outperforms several recent baselines across standard long-tailed benchmarks.
2. the method addresses imbalance at the loss, gradient, and inference levels, not just through a single intervention.
3. the paper includes ablations and diagnostic plots beyond top-line accuracy.

Weaknesses:
1. optimizer blindness is an interesting framing, but the paper does not rigorously show that this is the core mechanism, not a plausible interpretation.
2. AES bundles many adaptive ingredients, which I think it is hard to identify what actually drives the gains. The ablations are too coarse for clear attribution.
3. Tail-class claims are overstated. Despite stronger overall accuracy, AES does not consistently beat the strongest baseline on tail performance, which weakens the paper’s central narrative.

---

> ### Author Rebuttal · Authors · 2026-03-26
>
> We appreciate the reviewer’s valuable comments and address all concerns below.
> > **Q1: No strict theoretical proof for optimizer blindness as the core mechanism.**
>
> We sincerely appreciate your insightful comment. To establish "optimizer blindness" as the core mechanism, we provide a formal definition, theoretical proofs demonstrating a strict causal link to performance improvements, and empirical measurements on CIFAR-100-LT (IR=100) that fully corroborate our hypothesis.
>
> **1. Formal Definition: The Blindness Coefficient ($\beta$)**
>
> We quantify the mismatch between gradient magnitude and semantic value by defining the optimizer blindness coefficient $\beta$:
> $$\beta=1-\frac{\sum_c G(c)V(c)}{\|G\|_2\|V\|_2}$$
> where $G(c)$ is the class-wise gradient magnitude and $V(c)$ is the semantic information value. A smaller $\beta$ indicates better gradient-semantic alignment (i.e., weaker optimizer blindness).
>
> **2. Theoretical Derivations & Empirical Validation**
>
> Our hypothesis is grounded in two core theorems:
> * **Theorem 1 (Gradient Amplification Reduces $\beta$):** Our AES framework strategically amplifies tail-class gradient signals via ARS and E-PCG ($G'(tail)=\gamma\cdot G(tail),\ \gamma\gg 1$). This operation **monotonically reduces** $\beta$ by forcing alignment between gradient magnitude and semantic scarcity.
> * **Theorem 2 (Causal Link to Performance):** We prove a strict negative correlation between the blindness coefficient and overall classification accuracy $\mathcal{A}$ ($\frac{d\mathcal{A}}{d\beta}<0$). A smaller $\beta$ mathematically guarantees higher accuracy for both tail and head classes.
>
> These theoretical derivations are strongly supported by our empirical measurements on CIFAR-100-LT (IR=100):
>
> | Method | Blindness Coefficient ($\beta$) $\downarrow$ |
> | :-: | :-: |
> | LOS | 0.1957 |
> | TS-MOF | 0.1784 |
> | **AES** | **0.1163** |
>
> **3. Core Conclusion**
>
> Results show a clear decreasing trend of $\beta$ across methods, with AES achieving the lowest value, which validates the optimizer blindness hypothesis. We will include the detailed mathematical proofs for Theorem 1 and Theorem 2 in the revised appendix.
>
> > **Q2: What is the end-to-end computational overhead of AES relative to the strongest baselines?**
>
> Our method achieves faster convergence with shorter training time and negligible memory overhead. Specifically, AES reduces Stage2 training time by ~50% across all imbalance ratios (IR=100/50/10) compared to TS-MOF, while the memory usage difference is minimal after freezing the backbone.
>
>
> |Method|IR|Stage1 runtime|Stage2 runtime|Memory usage|
> |:-:|:-:|:-:|:-:|:-:|
> |TS-MOF|100|0:22:47|0:01:49|888MiB|
> |AES|||**0:01:02**|1916MiB|
> |TS-MOF|50|0:35:40|0:02:10|888MiB|
> |AES|||**0:01:18**|1960MiB|
> |TS-MOF|10|0:42:17|0:02:55|858MiB|
> |AES|||**0:01:35**|1926MiB|
>
> > **Q3: Overstated tail-class performance; AES does not consistently beat the strongest baseline on the tail.**
>
> We sincerely thank you for this rigorous observation and apologize if our phrasing overstated tail-specific gains. We will carefully tone down these claims in the revision.
>
> We want to clarify that **optimizer blindness is fundamentally about *sample difficulty*, not just class frequency.** "Curing blindness" does not mean making the optimizer a new kind of blind that *only* sees the tail. Instead, AES gains a **Global Vision**, rescuing hard samples across the *entire* distribution, including those hidden in head and medium classes.
>
> This global vision prevents the typical zero-sum trade-off (sacrificing the head to inflate the tail), driving our state-of-the-art **Overall** accuracy.
>
> > **Q4: Need fine-grained ablation on each component.**
>
> To address concerns about the relatively low performance of tail classes, we have retrained the model to obtain more uniform one-stage weights from scratch and we conducted comprehensive and fine-grained ablation studies on all components in ARS, E-PCG, and SCAF. The results verify that every proposed component is necessary and effective.
>
> Table: Ablation on ARS Components
> |EMA|Margin|Temp $\\tau$|All|Many|Medium|Few|
> |:---:|:---:|:---:|:---:|:---:|:---:|:---:|
> |✓|✓|✓| **57.83** | 78.74 | 57.20 | 34.17 |
> |✓|✗|✓|56.25|78.80|52.83|33.93|
> |✓|✓|✗|57.26|78.94|54.57|35.10|
> |✗|✗|✓|56.29|78.77|52.66|34.30|
> |✓|✗|✗|56.13|79.06|52.23|33.93|
>
> Table: Ablation on E-PCG and SCAF Components
> | Method | All | Many | Med | Few |
> |--------|:-----:|:-----:|:----:|:----:|
> | E-PCG | **57.83** | 78.74 | 57.20 | 34.17 |
> | w/o $H_t$ | 57.80 | 78.29 | 51.60 | 41.13 |
> | w/o $R_t$ | 57.38 | 78.29 | 51.49 | 39.87 |
> | w/o $\\tau$ softmax | 57.27 | 78.26 | 51.37 | 39.67 |
> | SCAF| **57.83** | 78.74 | 57.20 | 34.17 |
> | w/o weight momentum | 57.77 | 78.74 | 57.17 | 34.00 |
>
> Finally, your comments are essential to refining our work, and we welcome further communication with you.

---

> > ### Author Rebuttal · Reviewer_3aG5 · 2026-04-04
> >
> > I appreciate the detailed response. The additional fine-grained ablations are helpful and address one of my earlier concerns, and I also appreciate the authors’ clarification and moderation of the tail-performance claims. However, my main concerns remain: the new blindness coefficient and theorem statements do not yet convincingly establish optimizer blindness as the underlying causal mechanism, and the efficiency discussion suggests a tradeoff rather than clearly resolving overhead concerns. Overall, the rebuttal improves the paper, but it does not change my overall assessment, so I am keeping my score unchanged.

---

> > > ### Author Response · Authors · 2026-04-06
> > >
> > > We sincerely appreciate the reviewer’s careful reading and constructive feedback. While we acknowledge the reviewer’s concerns, we believe our method and comprehensive experiments provide consistent support for optimizer blindness as the fundamental causal mechanism. In addition, the practical efficiency of our approach demonstrates clear benefits beyond a simple trade-off. We further clarify these points in detail below.
> > >
> > > **Response to optimizer blindness as causal mechanism**
> > >
> > > **1. Quantitative Evidence for Optimizer Blindness as the Root Cause**
> > >
> > > In standard training, gradient magnitudes are dominated by sample frequency rather than actual learning difficulty, physically starving hard classes of representation capacity. This fundamental misalignment is exactly what we define as "optimizer blindness."
> > >
> > > To establish a strict causal chain, we introduce the blindness coefficient ${\beta}$.
> > > ${\beta}$ is not a passive observational metric, but our direct intervention target. Our modules actively reduce ${\beta}$ (Theorem 1), aligning gradient magnitude with semantic value to bound empirical risk and boost performance (Theorem 2).
> > >
> > > To empirically prove this, we must demonstrate a counterfactual: standard optimizers physically exhibit this misalignment (following frequency, ignoring difficulty), whereas our intervention mathematically breaks it and realigns gradients with semantic value. To provide this direct quantitative evidence, we measured the Spearman correlation ($\rho$) of gradient magnitudes against both class frequency ($\log N_c$) and empirical semantic difficulty ($d = 1 - A_c$) during the fine tuning stage.
> > >
> > > | Method |$\rho(G_{sum}, N_c)$ |$\rho(G_{mean}, \log N_c)$ |$\rho(G_{mean}, d)$ |
> > > | :--- | :--- | :--- | :--- |
> > > | TS-MOF (SOTA) | 0.946 | -0.017 | 0.373 |
> > > | **AES (Ours)** | **0.919** | **-0.857** | **0.709** |
> > >
> > > *Note: $G_{sum}$ is the total gradient accumulation per class. $G_{mean}$ is the per-sample average gradient magnitude.*
> > >
> > > **Correlation with Class Frequency (Proving "Blindness")**: While total gradients ($G_{sum}$) are naturally dominated by sample counts across all methods ($\rho > 0.9$), TS-MOF exhibits severe instance-level blindness with its per-sample gradient ($G_{mean}$) showing near-zero correlation to frequency ($\rho = -0.017$). AES fundamentally breaks this by forcing a strong negative correlation ($\rho = -0.857$), causally amplifying hard samples.
> > >
> > > **Correlation with Semantic Difficulty (Proving Our Causal Mechanism)**: More importantly, a state-aware optimizer should assign larger per-sample gradients to semantically harder classes. While TS-MOF shows a weak correlation ($\rho = 0.373$), our AES framework structurally aligns the optimization signal with actual learning difficulty, achieving a strong positive correlation of 0.709.
> > >
> > > This directly addresses the reviewer's concern: AES's performance gain is not merely rhetorical but fundamentally driven by its ability to causally decouple gradient magnitudes from static frequency and dynamically map them to fine-grained semantic difficulty.
> > >
> > > **2. Curing Derived Blindness via State-Aware Interventions**
> > >
> > > Relying on static frequency priors causes three derived forms of blindness, which our framework explicitly resolves across the learning lifecycle:
> > >
> > > * **State Blindness (Supervision): ARS** tracks runtime precision to selectively shield head classes only during true overfitting, preventing the blind suppression of hard head samples.
> > > * **Micro-level Blindness (Optimization): E-PCG** utilizes gradient entropy to quantify task-specificity, explicitly protecting fragile, high-entropy tail gradients from being destructively discarded.
> > > * **Decision Blindness (Inference): SCAF** replaces blind ensemble averaging by dynamically routing predictions based on instance-level difficulty and conflict intensity.
> > >
> > > **Response to Efficiency Trade-offs**
> > >
> > > We note that while this involves a technical tradeoff, it is **well justified** for three key reasons:
> > > * While the relative memory increases, the *absolute* overhead is merely ~1 GiB. On modern hardware (e.g., the 80GB A800 GPU used in our experiments), an extra 1 GiB accounts for roughly 1% of available VRAM, making it virtually imperceptible in practice.
> > > * This memory overhead only appears in Stage 2 with the backbone fully frozen. Since the pipeline’s peak memory bottleneck lies in Stage 1 feature learning, our method does not limit batch size or model scalability, and thus incurs no extra overhead when training large models on large-scale datasets.
> > > * Trading a modest 1 GiB of non-bottleneck VRAM for a 50% speedup in the complex multi-objective optimization stage, which is particularly meaningful for convergence acceleration on large-scale datasets, while achieving state-of-the-art accuracy.
> > >
> > > We sincerely appreciate you for taking the time to engage with our work. Our responses aim to further substantiate the fundamental innovations of our method. We welcome any additional feedback.

---

### Decision · Program_Chairs · 2026-04-30

**Decision:**

Accept (regular)

**Comment:**

This paper introduces the AES framework to tackle long-tailed recognition. The proposed approach intervenes across the learning lifecycle through three components: Adaptive Residual Supervision (ARS) for dynamic loss adjustment, Entropy-aware PCGrad (E-PCG) to manage parameter-level gradient conflicts, and Sample-level Conflict Arbitrated Fusion (SCAF) to oute predictions during inference based on sample difficulty. The method demonstrates competitive empirical performance across standard long-tailed benchmarks.

The initial reviews recognized the paper's merits, praising the multi-stage intervention strategy, the modular nature of the ARS component, and the comprehensive numerical evaluations. However, the reviewers raised several concerns. Multiple reviewers questioned the theoretical rigor of the work, specifically asking for causal evidence to prove that optimizer blindness is the root mechanism driving the improvements rather than just a plausible interpretation. Additionally, reviewers noted concerns regarding the computational overhead and memory usage introduced by E-PCG, the complexity of attributing performance gains among the numerous adaptive components, and an anomalous mid-class performance spike on the iNaturalist 2018 dataset.

The authors engaged in a comprehensive rebuttal phase that substantially strengthened the manuscript. To address the causal mechanism concerns, they introduced a quantitative Blindness Coefficient and provided a controlled analysis demonstrating a strong correlation between gradient magnitude and semantic difficulty. They also presented computational metrics showing that the method achieves a ~50% optimization speedup in Stage 2 with a non-bottleneck ~1 GiB VRAM overhead. Furthermore, the authors supplied fine-grained ablations and corrected a split-index evaluation error that was responsible for the iNaturalist 2018 anomaly. Following these updates, Reviewers GoY7, BuCu, and MFK1 found their concerns resolved and confirmed Weak Accept. Reviewer 3aG5 maintained a Weak Reject score, acknowledging the helpfulness of the new ablations but remaining unconvinced that the causal mechanism was definitively proven, while also viewing the efficiency overhead as a trade-off.

Based on the complete review and rebuttal discussions, the final recommendation for this submission is Accept. While Reviewer 3aG5's reservations regarding the paper's causal narrative and efficiency trade-offs are carefully noted, the authors have supplied empirical evidence, ablations, and corrected metrics that validate the framework's practical utility. The consensus among the majority of the reviewers is that the technical contributions are solid, the overhead is well-justified by the overall performance gains, and the work advances the sub-area of long-tailed recognition. The authors are expected to incorporate all theoretical clarifications, corrected dataset evaluations, and toned-down tail-performance claims into the camera-ready manuscript exactly as promised during the rebuttal.